# TRAINING INVARIANCES AND THE LOW-RANK PHENOMENON: BEYOND LINEAR NETWORKS

**Thien Le & Stefanie Jegelka**
Massachusetts Institute of Technology
`{thienle,stefje}@mit.edu`

## ABSTRACT

The implicit bias induced by the training of neural networks has become a topic of rigorous study. In the limit of gradient flow and gradient descent with appropriate step size, it has been shown that when one trains a deep linear network with logistic or exponential loss on linearly separable data, the weights converge to rank-1 matrices. In this paper, we extend this theoretical result to the last few linear layers of the much wider class of nonlinear ReLU-activated feedforward networks containing fully-connected layers and skip connections. Similar to the linear case, the proof relies on specific local training invariances, sometimes referred to as alignment, which we show to hold for submatrices where neurons are stably-activated in all training examples, and it reflects empirical results in the literature. We also show this is not true in general for the full matrix of ReLU fully-connected layers. Our proof relies on a specific decomposition of the network into a multilinear function and another ReLU network whose weights are constant under a certain parameter directional convergence.

## 1 INTRODUCTION

Recently, great progress has been made in understanding the trajectory of gradient flow (GF) (Ji & Telgarsky, 2019; 2020; Lyu & Li, 2020), gradient descent (GD) (Ji & Telgarsky, 2019; Arora et al., 2018) and stochastic gradient descent (SGD) (Neyshabur et al., 2015; 2017) in the training of neural networks. While good theory has been developed for deep linear networks (Zhou & Liang, 2018; Arora et al., 2018), practical architectures such as ReLU fully-connected networks or ResNets are highly non-linear. This causes the underlying optimization problem (usually empirical risk minimization) to be highly non-smooth (e.g. for ReLUs) and non-convex, necessitating special tools such as the Clarke subdifferential (Clarke, 1983).

One of the many exciting results of this body of literature is that gradient-based algorithms exhibit some form of implicit regularization: the optimization algorithm prefers some stationary points to others. In particular, a wide range of implicit biases has been shown in practice (Huh et al., 2021) and proven for deep linear networks (Arora et al., 2018; Ji & Telgarsky, 2019), convolutional neural networks (Gunasekar et al., 2018) and homogeneous networks (Ji & Telgarsky, 2020). One well known result for linearly separable data is that in various practical settings, linear networks converge to the solution of the hard SVM problem, i.e., a max-margin classifier (Ji & Telgarsky, 2019), while a relaxed version is true for CNNs (Gunasekar et al., 2018). This holds even when the margin does not explicitly appear in the optimization objective - hence the name implicit regularization.

Another strong form of implicit regularization relates to the structure of weight matrices of fully connected networks. In particular, Ji & Telgarsky (2019) prove that for deep *linear* networks for binary classification, weight matrices tend to rank-1 matrices in Frobenius norm as a result of GF/GD training, and that adjacent layers' singular vectors align. The max-margin phenomenon follows as a result. In practice, Huh et al. (2021) empirically document the low rank bias across different non-linear architectures. However, their results include ReLU fully-connected networks, CNNs and ResNet, which are not all covered by the existing theory. Beyond linear fully connected networks, Du et al. (2018) show vertex-wise invariances for fully-connected ReLU networks and invariances in Frobenius norm differences between layers for CNNs. Yet, despite the evidence in (Huh et al., 2021), it has been an open theoretical question how more detailed structural relations between layers,

which, e.g., imply the low-rank result, generalize to other, structured or local nonlinear and possibly non-homogeneous architectures, and how to even characterize these.

Hence, in this work, we take steps to addressing a wider set of architectures and invariances. First, we show a class of vertex and edge-wise quantities that remain invariant during gradient flow training. Applying these invariances to architectures containing fully connected, convolutional and residual blocks, arranged appropriately, we obtain invariances of the singular values in adjacent (weight) matrices or submatrices. Second, we argue that a matrix-wise invariance is not always true for general ReLU fully-connected layers. Third, we obtain low-rank results for arbitrary non-homogeneous networks whose last few layers contain linear fully-connected and linear ResNet blocks. To the best of our knowledge, this is the first time a low-rank phenomenon is proven rigorously for these architectures.

Our theoretical results offer explanations for empirical observations on more general architectures, and apply to the experiments in (Huh et al., 2021) for ResNet and CNNs. They also include the squared loss used there, in addition to the exponential or logistic loss used in most theoretical low-rank results. Moreover, our Theorem 2 gives an explanation for the "reduced alignment" phenomenon observed by Ji & Telgarsky (2019), where in experiments on AlexNet over CIFAR-10 the ratio $\|W\|_2/\|W\|_F$ converges to a value strictly less than 1 for some fully-connected layer $W$ towards the end of the network.

One challenge in the analysis is the non-smoothness of the networks and ensuring an "operational" chain rule. To cope with this setting, we use a specific decomposition of an arbitrary ReLU architecture into a multilinear function and a ReLU network with +1/-1 weights. This reduction holds in the *stable sign regime*, a certain convergence setting of the parameters. This regime is different from stable activations, and is implied, e.g., by directional convergence of the parameters to a vector with non-zero entries (Lemma 1). This construction may be of independent interest.

In short, we make the following contributions to analyzing implicit biases of general architectures:

- We show vertex and edge-wise weight invariances during training with gradient flow. Via these invariances, we prove that for architectures containing fully-connected layers, convolutional layers and residual blocks, when appropriately organized into matrices, adjacent matrices or submatrices of neurons with stable activation pattern have bounded singular value (Theorem 1).

- In the stable sign regime, we show a low-rank bias for arbitrary nonhomogeneous feedforward networks whose last few layers are a composition of linear fully-connected and linear ResNet variant blocks (Theorem 2). In particular, if the Frobenius norms of these layers diverge, then the ratio between their operator norm and Frobenius norm is bounded non-trivially by an expression fully specified by the architecture. To the best of our knowledge, this is the first time this type of bias is shown for nonlinear, nonhomogeneous networks.

- We prove our results via a decomposition that reduces arbitrarily structured feedforward networks with positively-homogeneous activation (e.g., ReLU) to a multilinear structure (Lemma 1).

## 1.1 RELATED WORKS

Decomposition of fully-connected neural networks into a multilinear part and a 0-1 part has been used by Choromanska et al. (2015); Kawaguchi (2016), but their formulation does not apply to trajectory studies. In Section 3, we give a detailed comparison between their approach and ours. Our decomposition makes use of the construction of a tree network that Khim & Loh (2019) use to analyze generalization of fully-connected networks. We describe their approach in Section 2 and show how we extend their construction to arbitrary feedforward networks. This construction makes up the first part of the proof of our decomposition lemma. The paper also makes use of path enumeration of neural nets, which overlaps with the path-norm literature of Neyshabur et al. (2015; 2017). The distinction is that we are studying classical gradient flow, as opposed to SGD (Neyshabur et al., 2015) or its variants (Neyshabur et al., 2017).

For linear networks, low-rank bias is proven for separable data and exponential-tailed loss in Ji & Telgarsky (2019). Du et al. (2018) give certain vertex-wise invariances for fully-connected ReLU networks and Frobenius norm difference invariances for CNNs. Compared to their results, ours are slightly stronger since we prove that our invariances hold for almost every time $t$ on the gradient

flow trajectory, thus allowing for the use of a Fundamental Theorem of Calculus and downstream analysis. Moreover, the set of invariances we show is strictly larger. Radhakrishnan et al. (2020) show negative results when generalizing the low-rank bias from (Ji & Telgarsky, 2019) to vector-valued neural networks. In our work, we only consider scalar-valued neural networks performing binary classification. For linearly inseparable but rank-1 or whitened data, a recent line of work from Ergen & Pilanci (2021) gives explicit close form optimal solution, which is both low rank and aligned, for the regularized objective. This was done for both the linear and ReLU neural networks. In our work, we focus on the properties of the network along the gradient flow trajectory.

## 2 PRELIMINARIES AND NOTATION

For an integer $k \in \mathbb{N}$, we write the set $[k] := \{1, 2, \ldots, k\}$. For vectors, we extend the sign function $\mathrm{sgn} : \mathbb{R} \to \{-1, 1\}$ coordinate-wise as $\mathrm{sgn} : (x_i)_{i \in [d]} \mapsto (\mathrm{sgn}(x_i))_{i \in [d]}$. For some (usually the canonical) basis $(e_i)_{i \in [n]}$ in some vector space $\mathbb{R}^n$, for all $x \in \mathbb{R}^n$ we use the notation $[x]_i = \langle x, e_i \rangle$ to denote the $i$-th coordinate of $x$.

**Clarke subdifferential, definability and nonsmooth analysis.** The analysis of non-smooth functions is central to our results. For a locally Lipschitz function $f : D \to \mathbb{R}$ with open domain $D$, there exists a set $D^c \subseteq D$ of full Lebesgue measure on which the derivative $\nabla f$ exists everywhere by Rademacher's theorem. As a result, calculus can usually be done over the *Clarke subdifferential* $\partial f(x) := \mathrm{CONV} \left\{ \lim_{i \to \infty} \nabla f(x_i) \mid x_i \in D^c, x_i \to x \right\}$ where CONV denotes the convex hull.

The Clarke subdifferential generalizes both the smooth derivative when $f \in C^1$ (continuously differentiable) and the convex subdifferential when $f$ is convex. However, it only admits a chain rule with an inclusion and not equality, which is though necessary for backpropagation in deep learning. We do not delve in too much depth into Clarke subdifferentials in this paper, but use it when we extend previous results that also use this framework. We refer to e.g. (Davis et al., 2020; Ji & Telgarsky, 2020; Bolte & Pauwels, 2020) for more details.

**Neural networks.** Consider a neural network $\nu : \mathbb{R}^d \to \mathbb{R}$. The computation graph of $\nu$ is a weighted directed graph $G = (V, E, w)$ with weight function $w : E \to \mathbb{R}$. For each neuron $v \in V$, let $\mathrm{IN}_v := \{u \in V : uv \in E\}$ and $\mathrm{OUT}_v := \{w \in V : vw \in E\}$ be the input and output neurons of $v$. Let $\{i_1, i_2, \ldots, i_d\} =: I \subset V$ and $O := \{o\} \subset V$ be the set of input and output neurons defined as $\mathrm{IN}(i) = \emptyset = \mathrm{OUT}(o), \forall i \in I$. Each neuron $v \in V \backslash I$ is equipped with a positively 1-homogeneous activation function $\sigma_v$ (such as the ReLU $x \mapsto \max(x, 0)$, leaky ReLU $x \mapsto \max(x, \alpha x)$ for some small positive $\alpha$, or the linear activation).

To avoid unnecessary brackets, we write $w_e := w(e)$ for some $e \in E$. We will also write $w \in \mathbb{R}^{\overline{E}}$, where $\overline{E}$ is the set of learnable weights, as the vector of learnable parameters. Let $P$ be a path in $G$, i.e., a set of edges in $E$ that forms a path. We write $v \in P$ for some $v \in V$ if there exists $u \in V$ such that $uv \in P$ or $vu \in P$. Let $\rho$ be the number of distinct paths from any $i \in I$ to $o \in O$. Let $\mathcal{P} := \{p_1, \ldots, p_\rho\}$ be the enumeration of these paths. For a path $p \in \mathcal{P}$ and an input $x$ to the neural network, denote by $x_p$ the coordinate of $x$ used in $p$.

Given a binary classification dataset $\{(x_i, y_i)\}_{i \in [n]}$ with $x_i \in \mathbb{R}^d, \|x_i\| \leq 1$ and $y_i \in \{-1, 1\}$, we minimize the empirical risk $\mathcal{R}(w) = \frac{1}{n} \sum_{i=1}^n \ell(y_i \nu(x_i)) = \frac{1}{n} \sum_{i=1}^n \ell(\nu(y_i x_i))$ with loss $\ell : \mathbb{R} \to \mathbb{R}$, using gradient flow $\frac{dw(t)}{dt} \in -\partial \mathcal{R}(w(t))$.

As we detail the architectures used in this paper, we recall that the activation of each neuron is still positively-homogeneous. The networks considered here are assumed to be bias-free.

**Definition 1** (Feedforward networks). *A neural net $\nu$ with graph $G$ is a* feedforward network *if $G$ is a directed acyclic graph (DAG).*

**Definition 2** (Fully-connected networks). *A feedforward network $\nu$ with graph $G$ is a* fully-connected network *if there exists a partition of $V$ into $V = (I \equiv V_1) \sqcup V_2 \sqcup \ldots \sqcup (V_{L+1} \equiv O)$ such that for all $u, v \in V, uv \in E$ iff there exists $i \in [L]$ such that $u \in V_i$ and $v \in V_{i+1}$.*

**Definition 3** (Tree networks). *A feedforward network $\nu$ with graph $G$ is a* tree network *if the underlying undirected graph $G$ is a tree (undirected acyclic graph).*

Examples of feedforwards networks include ResNet (He et al., 2016), DenseNet (Huang et al., 2017), CNNs (Fukushima, 1980; LeCun et al., 2015) and other fully-connected ReLU architectures.

For a fully-connected network $\nu$ with layer partition $V =: V_1 \sqcup \ldots \sqcup V_{L+1}$ where $L$ is the number of (hidden) layers, let $n_i := |V_i|$ be the number of neurons in the $i$-th layer and enumerate $V_i = \{v_{i,j}\}_{j \in [n_i]}$. Weights in this architecture can be organized into matrices $W^{[1]}, W^{[2]}, \ldots, W^{[L]}$ where $\mathbb{R}^{n_{i+1} \times n_i} \ni W^{[i]} = ((w_{v_{i,j} v_{i+1,k}}))_{j \in [n_i], k \in [n_{i+1}]}$, for all $i \in [L]$.

**Tree networks.** Most practical architectures are not tree networks, but trees have been used to prove generalization bounds for adversarial risk. In particular, for fully-connected neural networks $f$ whose activations are monotonically increasing and 1-Lipschitz, Khim & Loh (2019) define the *tree transform* as the tree network $Tf(x; w) = \sum_{p_L=1}^{n_L} W_{1,p_L}^{[L]} \sigma \left( \ldots \sum_{p_2=1}^{n_2} W_{p_3,p_2}^{[2]} \sigma \left( w_{p_2 \ldots p_L} + \sum_{p_1=1}^{n_1} W_{p_2,p_1}^{[1]} x_{p_1} \right) \right)$ for vectors $w$ with $\prod_{j=2}^{L} n_j$ entries, indexed by an $L$-tuple $(p_2, \ldots, p_L)$. We extend this idea in the next section.

## 3 STRUCTURAL LEMMA: DECOMPOSITION OF DEEP NETWORKS

We begin with a decomposition of a neural network into a multilinear and a non-weighted nonlinear part, which will greatly facilitate the chain rule that we need to apply in the analysis. Before stating the decomposition, we need the following definition of a path enumeration function, which computes the product of all weights and inputs on each path of a neural network.

**Definition 4** (Path enumeration function). *Let $\nu$ be a feedforward neural network with graph $G$ and paths $\mathcal{P} = \{p_1, \ldots, p_\rho\}$. The path enumeration function $h$ is defined for this network as $h : (x_1, x_2, \ldots, x_d) \mapsto \left( x_p \prod_{e \in p} w_e \right)_{p \in \mathcal{P}}$ where $x_p := x_k$ such that $i_k \in p$.*

We first state the main result of this section, proven in Appendix B.

**Lemma 1** (Decomposition). *Let $\nu : \mathbb{R}^d \to \mathbb{R}$ be a feedforward network with computation graph $G$, and $\rho$ the number of distinct maximal paths in $G$. Then there exists a tree network $\mu : \mathbb{R}^\rho \to \mathbb{R}$ such that $\nu = \mu \circ h$ where $h : \mathbb{R}^d \to \mathbb{R}^\rho$ is the path enumeration function of $G$. Furthermore, all weights in $\mu$ are either $-1$ or $+1$ and fully determined by the signs of the weights in $\nu$.*

**Path activation of ReLU networks in the literature.** The viewpoint that for every feedforward network $\nu$ there exists a tree network $\mu$ such that $\nu = \mu \circ h$ is not new and our emphasis here is on the fact that the description of $\mu : \mathbb{R}^\rho \to \mathbb{R}$ is fully determined by the signs of the weights. Indeed, in analyses of the loss landscape (Choromanska et al., 2015; Kawaguchi, 2016), ReLU networks are described as a sum over paths:

$$\nu(x; w) = \sum_{p \in \mathcal{P}} Z_p(x; w) \prod_{e \in p} w_e = \left\langle (Z_p(x; w))_{p \in \mathcal{P}}, h(x; w) \right\rangle_{\mathbb{R}^\rho}, \tag{1}$$

where $Z_p(x; w) = 1$ iff all ReLUs on path $p$ are active (have nonnegative preactivation) and 0 otherwise. One can then take $\mu$ as a tree network with no hidden layer, $\rho$ input neurons all connected to a single output neuron. However, this formulation complicates analyses of gradient trajectories, because of the explicit dependence of $Z_p$ on numerical values of $w$. In our lemma, $\mu$ is a tree network whose description depends only on the signs of the weights. If the weight signs (not necessarily the ReLU activation pattern!) are constant, $\mu$ is fixed, allowing for a chain rule to differentiate through it. That weight signs are constant is realistic, in the sense that it is implied by directional parameter convergence (Section 4.1). To see this, compare the partial derivative with respect to some $w_e$ (when it exists) between the two approaches, in the limit where weight signs are constant:

$$\text{(using Lemma 1)} \qquad \partial\nu/\partial w_e = \sum_{p \in \mathcal{P}|e \in p} [\nabla_w \mu(x; w)]_p \cdot x_p \prod_{f \in p, f \neq e} w_f, \tag{2}$$

$$\text{(using } Z_p \text{ in Eqn. 1)} \qquad \partial\nu/\partial w_e = \sum_{p \in \mathcal{P}|e \in p} \left( w_e \partial Z_p(x; w)/\partial w_e + Z_p(x; w) \right) \prod_{f \in p, f \neq e} w_f. \tag{3}$$

In particular, the dependence of Equation 2 on $w_e$ is extremely simple. The utility of this fact will be made precise in the next section when we study invariances.

**Proof sketch.** The proof contains two main steps. First, we "unroll" the feedforward network into a tree network that computes the same function by adding extra vertices, edges and enable weight sharing. This step is part of the tree transform in Khim & Loh (2019) if the neural network is a fully-connected network; we generalize it to work with arbitrary feedforward networks. Second, we "pull back" the weights towards the input nodes using positive homogeneity of the activations: $a \cdot \sigma(x) = \text{sgn}(a) \cdot \sigma(x|a|)$. This operation is first done on vertices closest to the output vertex (in number of edges on the unique path between any two vertices in a tree) and continues until all vertices have been processed. Finally, all the residual signs can be subsumed into $\mu$ by subdividing edges incident to input neurons. We give a quick illustration of the two steps described above for a fully-connected ReLU-activated network with 1 hidden layer in Appendix A.

## 4 Main Theorem: Training invariances

In this section, we put the previous decomposition lemma to use in proving an implicit regularization property of gradient flow when training deep neural networks.

### 4.1 Stable sign regime: a consequence of directional convergence

Recall the gradient flow curve $\{w(t)\}_{t \in [0,\infty)}$ defined by the differential inclusion $\frac{dw(t)}{dt} \in -\partial \mathcal{R}(w(t))$. We first state the main assumption in this section.

**Assumption 1** (Stable sign regime)**.** *For some $t_0 < t_N \in [0,\infty]$, we assume that for all $t \in [t_0, t_N), \text{sgn}(w(t)) = \text{sgn}(w(t_0))$. If this holds, we say that gradient flow is in a* stable sign regime. *Without loss of generality, when using this assumption, we identify $t_0$ with $0$ and write "for some $t \geq 0$" to mean "for some $t \in [t_0, t_N)$".*

In fact, the following assumption - the existence and finiteness part of which has been proven in (Ji & Telgarsky, 2020) for homogeneous networks, is sufficient.

**Assumption 2** (Directional convergence to non-vanishing limit in each entry)**.** *We assume that* $\frac{w(t)}{\|w(t)\|_2} \xrightarrow{t \to \infty} \overline{w}$ *exists, is finite in each entry and furthermore, for all $e \in E, \overline{w}_e \neq 0$.*

**Motivation and justification.** It is straightforward to see that Assumption 1 follows from Assumption 2 but we provide a proof in the Appendix (Claim 1). Directional convergence was proven by Ji & Telgarsky (2020) for the exponential/logistic loss and the class of homogeneous networks, under additional mild assumptions. This fact justifies the first part of Assumption 2 for these architectures. The second part of Assumption 2 is pathological for our case, in the sense that directional convergence alone does not imply stable signs (for example, a weight that converges to 0 can change sign an infinite number of times).

**Pointwise convergence is too strong in general.** Note also that assuming pointwise convergence of the weights (i.e. $\lim_{t \to \infty} w(t)$ exists and is finite) is a much stronger statement, which is not true for the case of exponential/logistic loss and homogeneous networks (since $\|w(t)\|_2$ diverges, see for example Lyu & Li (2020), Ji & Telgarsky (2020), Ji & Telgarsky (2019)). Even when pointwise convergence holds, it would immediately reduce statements on asymptotic properties of gradient flow on ReLU activated architectures to that of linearly activated architectures. One may want to assume that gradient flow starts in the final affine piece prior to its pointwise convergence and thus activation patterns are fixed throughout training and the behavior is (multi)linear. In contrast, directional convergence of the weights *does not* imply such a reduction from the ReLU-activation to the linear case. Similarly, with stable signs, the parts of the input where the network is linear are not convex, as opposed to the linearized case (Hanin & Rolnick, 2019) (see also Claim 2).

**Stable sign implication.** The motivation for Assumption 1 is that weights in the tree network $\mu$ in Lemma 1 are fully determined by the *signs* of the weights in the original feedforward network $\nu$. Thus, under Assumption 1, one can completely fix the weights of $\mu$ - it has no learnable parameters. Since we have the decomposition $\nu = \mu \circ h$ where $h$ is the path enumeration function, dynamics of $\mu$ are fully determined by dynamics of $h$ in the stable sign regime. To complete the picture, observe that $h$ is highly multilinear in structure: the degree of a particular edge weight $w_e$ in each entry of $h$ is at most 1 by definition of a path; and if $\nu$ is fully-connected, then $h$ is a $\mathbb{R}^{n_1 \times n_2 \times \ldots \times n_L}$ tensor.

## 4.2 TRAINING INVARIANCES

First, we state an assumption on the loss function that holds for most losses used in practice, such as the logistic, exponential or squared loss.

**Assumption 3** (Differentiable loss). *The loss function $\ell : \mathbb{R} \rightarrow \mathbb{R}$ is differentiable everywhere.*

**Lemma 2** (Vertex-wise invariance). *Under Assumptions 1, and 3, for all $v \in V \backslash \{I \cup O\}$ such that all edges incident to $v$ have learnable weights, for a.e. time $t \geq 0$,*

$$\sum_{u \in \text{IN}_v} w_{uv}^2(t) - \sum_{b \in \text{OUT}_v} w_{vb}^2(t) = \sum_{u \in \text{IN}_v} w_{uv}^2(0) - \sum_{b \in \text{OUT}_v} w_{vb}^2(0). \tag{4}$$

*If we also have $\text{IN}_u = \text{IN}_v = \text{IN}$ and $\text{OUT}_u = \text{OUT}_v = \text{OUT}$ and $u$ and $v$ have the same activation pattern (preactivation has the same sign) for each training example and for a.e time $t \geq 0$, then for a.e. time $t \geq 0$,*

$$\sum_{a \in \text{IN}} w_{au}(t)w_{av}(t) - \sum_{b \in \text{OUT}} w_{ub}(t)w_{vb}(t) = \sum_{a \in \text{IN}} w_{au}(0)w_{av}(0) - \sum_{b \in \text{OUT}} w_{ub}(0)w_{vb}(0). \tag{5}$$

**Comparison to Du et al. (2018)** A closely related form of Equation 4 in Lemma 2 has appeared in Du et al. (2018) (Theorem 2.1) for fully-connected ReLU/leaky-ReLU networks. In particular, the authors showed that the difference between incoming and outgoing weights does not change. Invoking the Fundamental Theorem of Calculus (FTC) over this statement will return ours. However, their proof may not hold on a nonnegligible set of time $t$ due to the use of the operational chain rule that holds only for almost all $w_e$. Thus the FTC can fail. The stronger form we showed here is useful in proving downstream algorithmic consequences, such as the low rank phenomenon. Furthermore, our result also holds for arbitrary feedforward architectures and not just the fully-connected case.

Before we put Lemma 2 to use, we list definitions of some ResNet variants.

**Definition 5.** *Denote ResNetIdentity, ResNetDiagonal and ResNetFree to be the version of ResNet described in He et al. (2016) where the residual block is defined respectively as*

1. *$r(x; U, Y) = \sigma(U\sigma(Yx) + Ix)$ where $x \in \mathbb{R}^a, Y, U^\top \in \mathbb{R}^{b \times a}$, and $I$ is the identity,*

2. *$r(x; U, Y, D) = \sigma(U\sigma(Yx) + Dx)$ where $x \in \mathbb{R}^a, Y, U^\top \in \mathbb{R}^{b \times a}$, and $D$ is diagonal,*

3. *$r(x; U, Y, Z) = \sigma(U\sigma(Yx) + Zx)$ where $x \in \mathbb{R}^a, Y \in \mathbb{R}^{b \times a}, U \in \mathbb{R}^{c \times b}$ and $Z \in \mathbb{R}^{c \times a}$.*

ResNetIdentity is the most common version of ResNet in practice. ResNetIdentity is a special case of ResNetDiagonal, which is a special case of ResNetFree. Yet, theorems for ResNetFree do not generalize trivially to the remaining variants, due to the restriction of Lemma 2 and Lemma 3 to vertices adjacent to all learnable weights and layers containing all learnable weights. For readability, we introduce the following notation:

**Definition 6** (Submatrices of active neurons). *Fix some time $t$, let $W \in R^{a \times b}$ be a weight matrix from some set of $a$ neurons to another set of $b$ neurons. Let $I_{active} \subseteq [b]$ be the set of $b$ neurons that are active (linear or ReLU with nonnegative preactivation). We write $[W^\top W]_{active} \in \mathbb{R}^{|I_{active}| \times |I_{active}|}$ for the submatrix of $W^\top W$ with rows and columns from $I_{active}$. Similarly, if $W' \in b \times c$ is another weight matrix from the same set of $b$ neurons to another set of $c$ neurons then $[W'W'^\top]_{active}$ is defined as the submatrix with rows and columns from $I_{active}$.*

When applying Lemma 2 to specific architectures, we obtain the following:

**Theorem 1** (Matrix-wise invariances). *Recall that a convolutional layer with number of input kernels $a$, kernel size $b$ and number of output kernels $c$ and is a tensor in $\mathbb{R}^{a \times b \times c}$. Under Assumptions 1 and 3, we have the following matrix-wise invariance for a.e. time $t \geq 0$:*

$$\frac{\mathrm{d}}{\mathrm{d}t}\left(\left[W_2(t)^\top W_2(t)\right]_{active} - \left[W_1(t)W_1(t)^\top\right]_{active}\right) = 0, \textit{for:} \tag{6}$$

1. *(Fully-connected layers) $W_1 \in \mathbb{R}^{b \times a}$ and $W_2 \in \mathbb{R}^{c \times b}$ consecutive fully-connected layers,*

2. *(Convolutional layers) $W_1$ is convolutional, viewed as a flattening to a matrix $\mathbb{R}^{c \times (a \times b)}$, and $W_2$ adjacent convolutional, viewed as a flattening to a matrix $\mathbb{R}^{(d \times e) \times c}$,*

3. *(Within residual block of ResNet) $W_1 = Y$ and $W_2 = U$ where $r(x; U, Y, Z)$ is a residual block of ResNetIdentity, ResNetDiagonal or ResNetFree,*

4. *(Between residual blocks of ResNet) $W_1 = \begin{bmatrix} U_1 & Z_1 \end{bmatrix}$, $W_2 = \begin{bmatrix} Y_2 \\ Z_2 \end{bmatrix}$ where $r(x; U_j, Y_j, Z_j)$, $j \in \{1, 2\}$ are consecutive ResNetFree blocks,*

5. *(Convolutional-fully-connected layers) $W_1$ convolutional, viewed as a flattening to a matrix $\mathbb{R}^{c \times (a \times b)}$ and $W_2$ adjacent fully-connected layer, viewed as an rearrangement to $\mathbb{R}^{d \times c}$,*

6. *(Convolutional-ResNetFree block) $W_1$ convolutional, viewed as a flattening to a matrix $\mathbb{R}^{c \times (a \times b)}$ and $W_2$ is a rearrangement of $\begin{bmatrix} U & Z \end{bmatrix}$ into an element of $\mathbb{R}^{d \times c}$, where $r(x; U, Y, Z)$ is an adjacent ResNetFree block,*

7. *(ResNetFree block-fully-connected layers and vice versa) $W_1 = \begin{bmatrix} U & Z \end{bmatrix} \in \mathbb{R}^{b \times a}$ and $W_2 \in \mathbb{R}^{c \times b}$ adjacent fully-connected or $W_1 \in \mathbb{R}^{b \times a}$ fully-connected and $W_2 = \begin{bmatrix} Y \\ Z \end{bmatrix} \in \mathbb{R}^{c \times b}$ adjacent ResNet block where $r(x; U, Y, Z)$ is the ResNetFree block.*

We emphasize that the above theorem only makes *local* requirements on the neural network, to have local parts that are either fully-connected, convolutional or a residual block. The only global architecture requirement is feedforward-ness. The first point of Theorem 1 admits an extremely simple proof for the linear fully-connected network case in Arora et al. (2018) (Theorem 1).

**Significance of Theorem 1.** If we have a set of neurons that is active throughout training (which is vacuously true for linear layers), we can invoke an FTC and get $W_2(t)^\top W_2(t) - W_1(t) W_1(t)^\top = W_2(0)^\top W_2(0) - W_1(0) W_1(0)^\top$ for the submatrix restricted to these neurons. Assume for simplicity that the right hand side is 0, then the singular values of $W_1$ and $W_2$ are the same for each of the cases listed in Theorem 1. If we can form a chain of matrices whose singular values are the same by iteratively invoking Theorem 1, then all matrices considered have the same singular values as the final fully-connected layer that connects to the output. Recall that our networks are scalar-valued, so the final layer is a row vector, which is rank 1 and thus all layers considered in the chain have rank 1, which is useful in the next section.

**Proof sketch of Theorem 1.** Given Lemma 2, we demonstrate the proof for the first point. The remaining points admit the exact same proof technique but on different matrices, which require some bookkeeping. Let $W_1 \in \mathbb{R}^{b \times a}$ and $W_2 \in \mathbb{R}^{c \times b}$ be two consecutive fully-connected layers for some $a, b, c \in \mathbb{N}$ number of vertices in these layers. Applying Equation 4 of Lemma 2 to each of the $b$ shared neurons between these two layers, one obtains the diagonal entries of Equation 6 of the Theorem. Now, apply Equation 5 to each pair among the $b$ shared neurons between these two layers to get the off-diagonal entries of Equation 6.

Next, we define layers for architectures where weights are not necessarily organized into matrices, e.g., ResNet or DenseNet.

**Definition 7** (Layer). *Let $F \subset E$ be such that 1) for all $e \neq f \in F$, there is no path that contains both $e$ and $f$; and 2) the graph $(V, E \setminus F, w)$ is disconnected. Then $F$ is called a* layer *of $G$.*

For this definition, we have the following invariance:

**Lemma 3** (Edge-wise invariance). *Under Assumptions 1 and 3, for all layers $F$ and $F'$ that contain all learnable weights, it holds that for a.e. time $t \geq 0$,*

$$\sum_{e \in F} w_e^2(t) - \sum_{f \in F'} w_f^2(t) = \sum_{e \in F} w_e^2(0) - \sum_{f \in F'} w_f^2(0). \tag{7}$$

**Significance of Lemma 3.** A flattening of a convolutional parameter tensor and a stacking of matrices in a ResNetDiagonal and ResNetFree block forms a layer. This lemma implies that the squared Frobenius norm of these matrices in the same network differs by a value that is fixed at initialization. The lemma also gives a direct implicit regularization for networks with biases, by treating neurons with bias as having an extra in-edge whose weight is the bias, from an extra in-vertex which is an input vertex with fixed input value 1.

### 4.3 PROOF SKETCH OF LEMMA 2 AND LEMMA 3

The proofs of Lemma 2, Lemma 3 and Theorem 1 share the technique of double counting paths, which we explain next. For simplicity, we assume here that we are working with a network that is differentiable everywhere in some domain that we are considering – we give a full general proof in the Appendix. The main proof idea was used in (Arora et al., 2018) and involves simply writing down the partial derivative of the risk. We have, for some particular weight $w_e, e \in E$, via the smooth chain rule

$$\frac{\partial \mathcal{R}(w)}{\partial w_e} = \frac{1}{n} \sum_{i=1}^{n} l'(y_i \nu(x_i; w)) \cdot y_i \cdot \frac{\partial \nu(w)}{\partial w_e} \tag{8}$$

$$= \frac{1}{n} \sum_{i=1}^{n} l'(y_i \nu(x_i; w)) \cdot y_i \cdot \sum_{p \in \mathcal{P}, p \ni e} \left[ \mu'(x_i; w) \right]_p \cdot (x_i)_p \prod_{f \in p, f \neq e} w_f, \tag{9}$$

where in the second line, we invoke the decomposition Lemma 1 and emphasize that $\mu$ has no learnable parameters in the stable sign regime. Now multiply $w_e$ to the above expression to get

$$w_e \frac{\partial \mathcal{R}(w)}{\partial w_e} = \sum_{p \in \mathcal{P}, p \ni e} \frac{1}{n} \sum_{i=1}^{n} A_{i,p}(w), \tag{10}$$

where $A_{i,p}(w) = l'(y_i \nu(x_i; w)) \cdot y_i \cdot \left[ \mu'(x_i; w) \right]_p \cdot (x_i)_p \prod_{f \in p} |w_f|$. Notice that $A_{i,p}(w)$ does not depend explicitly on the edge $e$, with respect to which we are differentiating (only through $w$). Thus, we sum over in-edges and out-edges of a particular $v$ satisfying the assumption of Lemma 2 to get

$$\sum_{u \in \text{IN}_v} w_{uv} \frac{\partial \mathcal{R}(w)}{\partial w_{uv}} = \sum_{p \in \mathcal{P}, p \ni v} \frac{1}{n} \sum_{i=1}^{n} A_{i,p}(w) = \sum_{b \in \text{OUT}_v} w_{vb} \frac{\partial \mathcal{R}(w)}{\partial w_{vb}}. \tag{11}$$

Note that the only difference between Equations 10 and 11 is the set of paths that we are summing over, and we double count this set of paths. We use the definition of gradient flow to obtain $\partial \mathcal{R}(w)/\partial w_e = \mathrm{d}w_e(t)/\mathrm{d}t$ and integrate with respect to time using a FTC to get the first part of Lemma 2. More work is needed to get the second part of Lemma 2, which is detailed in Appendix C. Finally, to get Lemma 3, we double count the set of all paths $\mathcal{P}$.

### 4.4 NONINVARIANCE OF GENERAL ReLU LAYERS

The restriction of Theorem 1 to submatrices of active neurons may appear limiting, but does not extend to the general case. With the same technique as above, we can write down the gradient for the Gram matrix $W_1^\top W_1$ for ReLU layers and show that it is not equal to its counterpart $W_2 W_2^\top$, thus giving a negative result:

**Lemma 4** (Noninvariance in ReLU layers). *Even under Assumptions 3 and 1, for a.e. time $t \geq 0$,*

$$\frac{\mathrm{d}}{\mathrm{d}t} \left( W_2(t)^\top W_2(t) - W_1(t) W_1(t)^\top \right) \neq 0, \tag{12}$$

*for the different pairs of $W_1, W_2$ detailed in Theorem 1.*

Despite the negative result, the closed form of the gradient for the Gram matrices can be shown to be low rank with another subgradient model. Details may be found in Appendix C.

## 5 CONSEQUENCES: LOW RANK PHENOMENON FOR NONLINEAR NONHOMOGENEOUS DEEP FEEDFORWARD NET

We apply the results from previous parts to prove a low-rank bias result for a large class of feedforward networks. To the best of our knowledge, this is the first time such a result is shown for this class of deep networks, although the linear fully-connected network analogue has been known for some time. In light of Theorem 1, we define a matrix representation of a layer:

**Definition 8** (Matrix representation of a layer). *The matrix representation for a ResNetFree block $r(x; U, Y, Z)$ is $\begin{bmatrix} U & Z \end{bmatrix}$; for a $T^{a,b,c} \in \mathbb{R}^{a \times b \times c}$ convolutional tensor it is the flattening to an element of $\mathbb{R}^{a \times (b \times c)}$; and for a fully-connected layer it is the weight matrix itself.*

**Theorem 2** (Reduced alignment for non-homogeneous networks). *Under Assumptions 1 and 3, let $\nu$ consist of an arbitrary feedforward neural network $\eta$, followed by $K \geq 0$ linear convolutional layers $(T^{a_k, b_k, c_k})_{k \in [K]}$, followed by $M \geq 0$ layers that are either linear ResNetFree blocks or linear fully-connected layers; and finally ending with a linear fully-connected layer $Fin$. For $j \in [K + M]$, denote by $W^{[j]}$ the matrix representation of the $j$-th layer after $\eta$, $N_r(j)$ the number of ResNetFree blocks between $j$ and $Fin$ exclusively and $V_c(j) := \max \dim W^{[M+1]} \cdot \prod_{j < k \leq M} \min(a_k, b_k)$ if $j \leq M$ and $1$ otherwise. Then there exists a constant $D \geq 0$ fixed at initialization such that for a.e. time $t > 0$,*

$$\frac{1}{8^{N_r(j)} V_c(j)} \|W^{[j]}(t)\|_F^2 - \|W^{[j]}(t)\|_2^2 \leq D, \ \forall j \in [K + M]. \tag{13}$$

*Furthermore, assume that $\|W^{[j]}\|_F \to \infty$ for some $j \in [K + M]$, then we have, as $t \to \infty$:*

$$1/\min\left(\mathrm{rank}(W^{[k]}), 8^{N_r(j)} V_c(j)\right) \leq \|W^{[k]}(t)\|_2^2 / \|W^{[k]}(t)\|_F^2 \leq 1, \tag{14}$$

*In particular, for the last few fully-connected layers $j$ with $N_r(j) = 0$ and $V_c(j) = 1$, we have:*

$$\left\| \frac{W^{[j]}(t)}{\|W^{[j]}(t)\|_F} - u_j(t) v_j^\top(t) \right\|_F \xrightarrow{t \to \infty} 0, \forall k \in [M], \qquad |\langle v_{j+1}, u_j \rangle| \xrightarrow{t \to \infty} 1, \tag{15}$$

*where $u_j$ and $v_j$ are the left and right principal singular vectors of $W^{[j]}$.*

**Corollary 1.** *For fully-connected networks with ReLU activations where the last $K$ layers are linear layers, trained with linearly separable data under logistic loss $\ell$, under the assumptions that $\mathcal{R}(w(0)) < \ell(0)$ and the limiting direction of weight vector (which exists (Ji & Telgarsky, 2020)) has no $0$ entries, Equation 15 holds for the last $K$ layers.*

**Significance** Equation 13 and its limiting counterpart Equation 14 quantify a low-rank phenomenon by providing a lower bound on the ratio of the largest squared singular value (the operator norm) and the sum of all squared singular values (the Frobenius norm). This lower bound depends on the *number* (not dimensions) of ResNetFree layers ($N_r$) and *certain dimensions* of convolutional layers ($V_c$). When the dimensions of ResNetFree layers are large, $\max \dim W^{[M+1]}$ is small and the number of input channels of convolutional layers are large, this lower bound is strictly better than the trivial lower bound of $1/\mathrm{rank}(W^{[k]})$. This is a quantification of the *reduced alignment* observed in (Ji & Telgarsky, 2019). In particular, for the last few fully connected layers (Equation 15, Corollary 1), the lower bound matches the upper bound of $1$ in the limit of Frobenius norm tending to infinity and the limiting weight matrices have rank 1 and adjacent layers align.

## 6 CONCLUDING REMARKS AND FUTURE DIRECTIONS

In this paper, we extend the proof of the low rank phenomenon, which has been widely observed in practice, beyond the linear network case. In particular, we address a variety of nonlinear architectural structures, homogeneous and non-homogeneous, which in this context have not been addressed theoretically before. To this end, we decomposed a feedforward ReLU/linear activated network into a composition of a multilinear function with a tree network. If the weights converge in direction to a vector with non-zero entries, the tree net is eventually fixed, allowing for chain rules to differentiate through. This leads to various matrix-wise invariances between fully-connected, convolution layers and ResNet blocks, enabling us to control the singular values of consecutive layers. In the end, we obtain a low-rank theorem for said local architectures.

Proving convergence to the stable sign regime for a wider set of architectures will strengthen Theorem 2. Another direction is to connect our low-rank bias results to the max-margin implicit regularization literature, which has been shown for linear networks and, more recently, certain 2-homogeneous architectures (Ji & Telgarsky, 2020).

### ACKNOWLEDGMENTS

This work was partially funded by NSF CAREER award 1553284 and NSF award 2134108. The authors thank the anonymous reviewers for their insightful feedback. We would also like to thank Matus Telgarsky for fruitful discussions on their related papers and on the Clarke subdifferential, and Kaifeng Lyu for pointing out an error in an earlier version of this paper.

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

## A ILLUSTRATION OF LEMMA 1

We give a quick illustration of the two steps described in Section 3 for a fully-connected ReLU-activated network with 1 hidden layer. Figure 1 describes the unrolling of the neural network (Figure 1a) into a tree network (Figure 1b). Figure 2a describes the weight pull-back in the hidden layer and Figure 2b describes the weight pull-back in the input layer. It is clear that the inputs of the tree net in Figure 2b are coordinates of the path enumeration function $h(x; w)$ in this example. Furthermore, weights in the tree net depend entirely on the signs of the original weights. The rest of the proof argues this intuition for general feed-forward neural nets. As a remark, in general, $\rho$ is a very large number - exponential in the number of layers for a fully-connected net with fixed width.

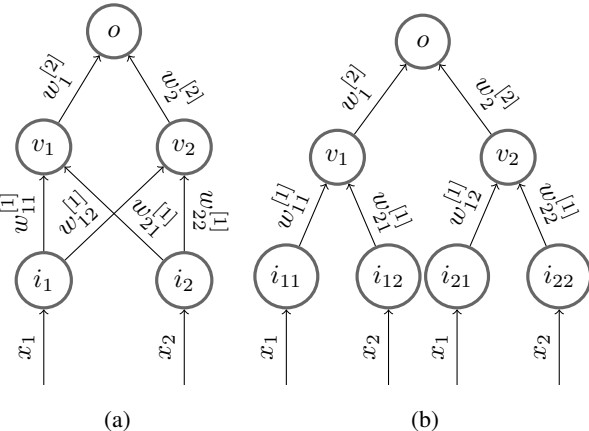

(a)                                              (b)

Figure 1: Transformation of a feedforward network into a tree net. All nodes apart from the input nodes use ReLU activation. The two neural nets drawn here compute the same function. This idea has been used in Khim & Loh (2019) to prove generalization bounds for adversarial risk.
.

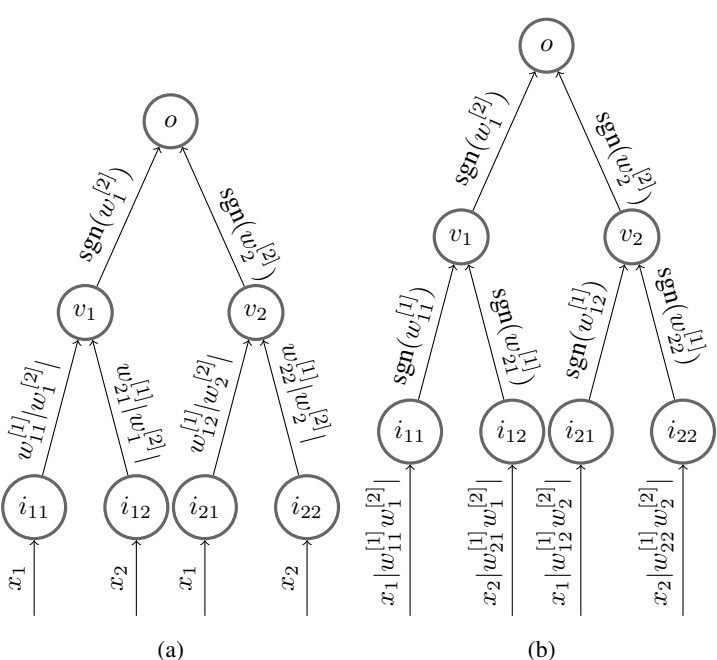

(a)                                              (b)

Figure 2: Pulling back of weights in a tree net. All nodes apart from the input nodes use the ReLU activation. The original net was drawn in Figure 1a and 1b. This is possible due to the positive-homogeneity of the activation function. From Figure 2b, one can recover the final tree net in Theorem 1 with weights from $\{-1, +1\}$ and input from the path enumeration function $h$ of the neural net by subdividing the edge incident to the input neurons and assign weights corresponding to $\text{sgn}(h(x))$.

## B  PROOF OF LEMMA 1

We first prove an absolute-valued version of Lemma 1 and show that the extension to Lemma 1 is straightforward. In other words, we first prove

**Lemma 5** (Absolute-valued decomposition). *For an arbitrary feed-forward neural network $\nu : \mathbb{R}^d \to \mathbb{R}$, there exists a tree network $\mu' : \mathbb{R}^\rho \to \mathbb{R}$ such that $\nu = \mu' \circ h'$ where $h'$ is the*

*absolute-valued path enumeration function defined as* $h'(x) = \left(x_p \prod_{e \in p} |w_e|\right)_{p \in \mathcal{P}}$. *Furthermore, the weights of $\mu'$ is in $\{-1, 1\}$ and only depends on the sign of the weights of the original network $\nu$.*

We first give some extra notation for this section. In general, we define:

- $\wp(S)$ to be the finite power set of some finite set $S$.
- $|p| \in \mathbb{N}_{\geq 0}$ to be the cardinality of a set $p$. When $p$ is a path then it is viewed as the number of edges.

To make it clear which graph we are referring to, define for an arbitrary feedforward neural network $\nu : \mathbb{R}^d \to \mathbb{R}$ with computation graph $G = (G[V], G[E], G[w])$:

- $I_G = \{i_1^G, \ldots, i_d^G\}$ to be the set of input node of $G$ and $O_G = \{o_G\}$ to be the set of the single output node of $G$.
- $\mathcal{P}_G$ to be the enumeration of all paths from any input node in $I_G$ to the output node $o_G$.
- $\mathcal{H}_G$ to be the enumeration of all paths from *any* node $v \in G[V]$ to the output node $o_G$. Note that if $G$ has more than 2 vertices then $\mathcal{H}_G \supset \mathcal{P}_G$.
- Each $v \in G[V]$ to be equipped with a fix activation $\sigma_v$ that is positively-1-homogeneous. To be precise, $G$ is enforced to be a DAG, is connected and is simple (no self-loop, at most 1 edge between any pair of vertices). Each node $v$ of $G[V]$ is equipped with: an activation function $G[\sigma](v)$ that is positively-1-homogeneous; a pre-activation function defined as:

$$(G[\text{PRE}](v))(x) = \begin{cases} x_j & \text{if } v \equiv i_j^G \text{ for some } j \in [d] \\ \sum_{u \in \text{IN}_v} \text{POST}_v(x) w_{uv} & \text{otherwise;} \end{cases} \quad (16)$$

and a post-activation function defined as

$$(G[\text{POST}](v))(x) = \sigma_v((G[\text{PRE}](v))(x)) \quad (17)$$

Note that $\text{POST}_{o_G} = \nu$.
- $v_p^G, e_p^G$ to be the vertex and edge, respectively, furthest from $o_G$ on some path $p$ of $G$.
- $x_p^G$ to be the unique $x_j$ such that $i_j^G \in p$.

**Definition 9** (Hasse diagram of inclusion). *Let $S$ be a finite set and a set $\mathcal{S} \subset \wp(S)$ of elements in $S$. A Hasse diagram of $\mathcal{S}$ is a directed unweighted graph $\text{HASSE}(\mathcal{S}) = (\mathcal{S}, E)$ where for any $p, q \in \mathcal{S}$, $pq \in E$ iff $p \subseteq q$ and $|q| - |p| = 1$.*

**Definition 10** (Unrolling of feedforward neural networks). *The unrolled tree neural network $\tau_G$ of $G$ is a tree neural network with computation graph $T_G$ where*

- *Unweighted graph $(T_G[V], T_G[E]) = \text{HASSE}(\mathcal{H}_G)$. In particular $T_G[V] = \mathcal{H}_G$ and we identify vertices of $T_G$ with paths in $G$.*
- *Weight function $T_G[w] : T_G[E] \ni pq \mapsto G[w](e_p)$.*
- *Activation $T_G[\sigma](v) := G[\sigma](v_p)$.*

**Lemma 6** (Unrolled network computes the same function). *Fix an arbitrary feedforward neural network $\nu : \mathbb{R}^d \to \mathbb{R}$ with computation graph $G$. Let $\tau_G$ be the unrolled tree network of $G$ with computation graph $T_G$. Then $\nu = \tau_G$.*

*Proof.* We proceed with induction on $\alpha_G := \max_{p \in \mathcal{P}} |p|$ the longest path between any input node and the output node of $G$. In the base case, set $\alpha_G = 0$. Then $V_G = \{o_G\}$ is a singleton and the neural network $\nu$ computes the activation of the input and return it. $\mathcal{H}_G = V_T = \{p_0\}$ then is a singleton containing just the trivial path that has just the output vertex of $G$ and no edges. The activation function attached to $p_0$ in $T$, by construction, is the activation of $o_G$. Thus, $\tau$ also simply returns the activation of the input.

Assume that $\tau_G = \nu$ for any $\nu$ with graph $G$ such that $\alpha_G \leq t - 1$ for some $t \geq 1$; and for the $\tau$ constructed as described. We will show that the proposition is also true when $\nu$ has graph $G$ with $\alpha_G = t$. Fix such a $\nu$ and $G$ that $\alpha_G = t$. We prove this induction step by:

1. First constructing $G'$ from $G$ such that $\nu' = \nu$ where $\nu'$ is the neural network computed by $G'$.

2. Then showing that $T_G = G'$ by constructing an isomorphism $\pi : G'[V] \rightarrow T_G[V]$ that preserves $E, w$ and $\sigma$.

**The construction of $G'$**  An illustration of the following steps can be found in Figure 3. Recall that $\text{IN}_{o_G} = \{v_1, \ldots, v_m\}$ is the set of in-vertices of $o_G$.

1. Create $m$ distinct, identical copies of $G$: $G_1, \ldots, G_m$.

2. For each $j \in [m]$, remove from $G_j$ all vertices $u$ (and their adjacent edges) such that there are no directed path from $u$ to $v_j$.

3. We now note that $\alpha_{G_j} \leq t - 1$ (to be argued) and invoke inductive hypothesis over $G_j$ to get an unrolled tree network $\tau_j$ with graph $T_j$ such that $\tau_j = \nu_j$ where $\nu_j$ is the neural network computed by $G_j$.

4. Finally, construct $G'$ by creating a new output vertex $o_{G'}$ and connect it to the output vertices $o_{T_j}$ for all $j \in [m]$. As a sanity check, since each $T_j$ is a tree network, so is $G'$. More precisely,

    (a) $G'[V] = \{o_{G'}\} \cup \bigcup_{j=1}^m T_j[V]$;
    (b) $G'[E] = \{o_{G_j} o_{G'} \mid j \in [m]\} \cup \bigcup_{j=1}^m T_j[E]$
    (c) $G'[w](e) = G[w](v_j o_G)$ if $e \equiv o_{G_j} o_{G'}$ for some $j \in [m]$ and $T_j[w](e)$, where $e \in T_j[E]$, otherwise.
    (d) $G'[\sigma](v) = G[\sigma](o_G)$ if $v \equiv o_G$ and $T_j[\sigma](v)$, where $v \in T_j[V]$, otherwise.

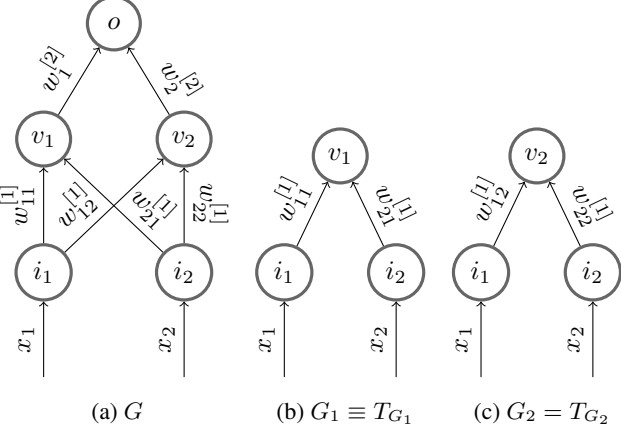

(a) $G$  (b) $G_1 \equiv T_{G_1}$  (c) $G_2 = T_{G_2}$

Figure 3: Construction of $G'$. $G_1$ and $G_2$ are the modified copies of $G$ in step 2. In step 3, the transformation $T_{G_i}$ happens to coincide with $G_i$ for $i = 1, 2$ in this case. $G'$ is created in step 4 by adding an extra vertex $o_{G'}$ and connecting it to $v_1$ and $v_2$ with the appropriate weights and activation and can be seen in Figure 1b.
.

**$G'$ is well-defined**  We verify each steps in the above construction:

1. This step is well-defined.

2. For any $G_j$, as long as $\alpha_G \geq 2$ (by definition), we always remove $o_G$ from each $G_j$ in each step; since $v_j$ is an in-vertex of $o_G$ and $G$ is a DAG. Otherwise, this step is well-defined.

3. Fix a $j \in [m]$. By construction (and since we always remove $o_G$ from $G_j$ in the previous step), $O_{G_j} = \{v_j\}$. If there is a path $p^*$ with length at least $t$ in $G_j$, then since $G_j[E] \subseteq G[E]$ and $o_G \in G[V] \setminus G_j[V]$, the path $p^* \cup \{o_G\}$ created by appending $o_G$ to $p^*$ is a valid path with length $t+1$. This violates the assumption that $\alpha_G = t$ and we conclude, by contradiction, that $\alpha_{G_j} \leq t-1$. This justifies the invocation of inductive hypothesis for $\nu_j$ to get a tree neural net $\tau_j$.

4. The final step is well-defined.

$\nu'$ **computes the same function as** $\nu$    Recall that $\nu'$ is the neural network with graph $G'$. We have for any input $x \in \mathbb{R}^d$

$$\nu'(x) = (G'[\text{POST}](o_{G'}))(x) \tag{18}$$

$$= G'[\sigma](o_{G'}) \left( \sum_{j=1}^{m} G'[w](v_j o_{G'}) \cdot (G'[\text{POST}](v_j))(x) \right) \tag{19}$$

$$= G[\sigma](o_G) \left( \sum_{j=1}^{m} G[w](v_j o_G) \cdot (T_j[\text{POST}](v_j))(x) \right) \tag{20}$$

$$= G[\sigma](o_G) \left( \sum_{j=1}^{m} G[w](v_j o_G) \cdot (G[\text{POST}](v_j))(x) \right) = \nu(x) \tag{21}$$

where we invoked the inductive hypothesis in the last line to get

$$(T_j[\text{POST}](v_j))(x) = \tau_j(x) = \nu_j(x) = (G[\text{POST}](v_j))(x), \tag{22}$$

and the rest are definitions.

$G'$ **is isomorphic to** $T_G$    Although this should be straightforward from the construction, we give a formal proof. Consider the isomorphism $\pi : G'[V] \to T_G[V]$ given as

$$\pi(v) = \begin{cases} o_{T_G} = \{o_G\} \in \mathcal{H}_G & \text{if } v \equiv o_{G'} \\ p \cup \{o_G\} & \text{if } T_j[V] \ni v \equiv p \in \mathcal{H}_{G_j} \text{ for some } j \in [m], \end{cases} \tag{23}$$

where paths are viewed as set of vertices. The second case is well-defined since $p \cup \{o_G\} \in \mathcal{H}_G$ for any $p \in \mathcal{H}_{G_j}$ for any $j \in [m]$ by construction of $G_j$.

We can now verify that $\pi$ is an isomorphism between the two structures. Fix $pq \in G'[E]$. Consider two separate cases: $pq \in T_j[E]$ for some $j$ and $pq \notin T_j[E]$ for any $j$. In the first case, by definition of $T_j$ as a Hasse diagram, $p = \{v_p\} \cup q$ are paths in $\mathcal{H}_{G_j}$. Thus, $\pi(p)\pi(q) = (p \cup \{o_G\})(q \cup \{o_G\})$ satisfying $\pi(p) = \{v_p\} \cup \pi(q)$. Thus, by definition of Hasse diagram, $\pi(p)\pi(q) \in T_G[E]$. Furthermore, $G'[w](pq) = G[w](e_p) = T_G[w](\pi(p)\pi(q))$ In the second case, we have $pq = v_j o_G$ for some $j \in [m]$. Thus, $\pi(p)\pi(q) = (\{v_j, o_G\})(\{o_G\}) \in T_G[E]$ also by definition of Hasse diagram. At the same time, $G'[w](pq) = G[w](v_j o_G) = T_G[w](\pi(p)\pi(q))$ by definition.

Fix $v \in G'[V]$. If $v = o_{G'}$ then $G'[\sigma](v) = G[\sigma](o_G)$. At the same time, $T_G[\sigma](\pi(v)) = G[\sigma](v_{\{o_G\}} = G[\sigma](o_G) = G'[\sigma](v)$. If $v \not\equiv o_{G'}$ then there is a $j \in [m]$ and a $p \in \mathcal{H}_{G_j}$ such that $v = p \in T_j[V]$. Then by definition,

$$G'[\sigma](v) = T_j[\sigma](v) = G[\sigma](v_p) = G[\sigma](v_p \cup \{o_{G'}\}) = T_G[\sigma](\pi(v)). \tag{24}$$

This completes the inductive proof that shows $\tau_G = \nu' = \nu$ for $G$ with $\alpha_G = t$. By mathematical induction, the claim holds for all neural network $\nu$. $\square$

**Lemma 7** (Pull-back of numerical weights). *Fix an arbitrary feedforward neural network $\nu : \mathbb{R}^d \to \mathbb{R}$ with computation graph $G$. Let $v \in G[V] \setminus (I_G \cup O_G)$ be an inner vertex of $G$ with $k$ in-edges $e_1, \ldots, e_k \in G[E]$ and a single out-edge $f \in G[E]$. Then the network $\nu'$ with computation graph $G' = (G[V], G[E], G'[w])$ defined as*

$$G'[w] : e \mapsto \begin{cases} G[w](e)|G[w](f)| & \text{if } e \equiv e_j \text{ for some } j \in [k] \\ \text{sgn}(G[w](f)) & \text{if } e \equiv f \\ G[w](e) & \text{otherwise.} \end{cases} \tag{25}$$

*computes the same function as $\nu$. In other words, we can pull the numerical values of $G[w](f)$ through $v$, into its in-edges; leaving behind only its sign.*

*When fixing input $x$ to $G$, one can extend this operation to $i_j \in I_G$ for $j \in [d]$. By setting $G'[w](f) = \text{sgn}(G[w](f))$ and update $x_j$ to $x_j|G[w](f)|$.*

*Proof.* Let the single out-vertex of $v$ be $b$. If suffices to show that $G[\text{PRE}](b) = G'[\text{PRE}](b)$. Fix an input $x$ to $\nu$. Let the $k$ in-edges of $v$ be $a_1, \ldots, a_k$. Since we only change edges incident to $v$, $G[\text{POST}]_{a_j} = G'[\text{POST}]_{a_j}$ for all $j \in [k]$. We have:

$$G'[\text{PRE}](b) = \text{sgn}(G[w](vb)) \cdot (G[\sigma](v))(G'[\text{PRE}](v)) \tag{26}$$

$$= \text{sgn}(G[w](vb)) \cdot (G[\sigma](v)) \left( \sum_{j=1}^{k} |G[w](vb)|G[w](a_jv) \cdot G[\text{POST}](a_j) \right) \tag{27}$$

$$= \text{sgn}(G[w](vb))|G[w](vb)| \cdot (G[\sigma](v)) \left( \sum_{j=1}^{k} G[w](a_jv) \cdot G[\text{POST}](a_j) \right) \tag{28}$$

$$= G[w](vb) \cdot (G[\sigma](v)) \left( \sum_{j=1}^{k} G[w](a_jv) \cdot G[\text{POST}](a_j) \right) = G[\text{PRE}](b), \tag{29}$$

where equation 28 is due to positive homogeneity of $\sigma_v$ and the rest are just definitions. $\square$

We can now give the proof of Lemma 5.

*Proof of Lemma 5.* Given an arbitrary feedforward neural network $\nu$ with computation graph $G = (G[V], G[E], G[w])$, we use Lemma 6 and get the unrolled tree network $T_G = (T_G[V], T_G[E], T_G[w])$ such that $\nu = \tau_G$.

Let $\pi = \pi_1, \pi_2, \ldots, \pi_\xi$ be an ordering of $T_G[V] \setminus O_{T_G}$ (so $\xi = |\mathcal{H}| - 1$) by a breadth first search on $T_G$ starting from $o_{T_G}$. In other words, if $d_{\text{topo}}(u, o_{T_G}) > d_{\text{topo}}(v, o_{T_G})$ then $u$ appears after $v$ in $\pi$, where $d_{\text{topo}}(a, b)$ is the number of edges on the unique path from $a$ to $b$ for some $a, b \in T_G[V]$. Iteratively apply Lemma 7 to vertex $\pi_1, \ldots, \pi_\xi$ in $T_G$ while maintaining the same function. After $\xi$ such applications, we arrive at a network $\mu'_G$ with graph $M_G$ such that $\mu'_G = \tau_G = \nu$. Recall that the pull-back operation of Lemma 7 only changes the tree weights. Furthermore, the $\pi$ ordering is chosen so that subsequent weight pull-back does not affect edges closer to $o_{T_G}$. Therefore, at iteration $j$,

1. $M_G[w](\pi_k q) = \text{sgn}(G[w](e_{\pi_k}))$ for all $k \leq j$, for some $q \in M_G[V]$ such that $(\pi_k)q \in M_G[E]$.

2. if $\pi_j \notin I_{M_G}$ then $M_G[w](r(\pi_j)) = G[w](e_r) \prod_{f \in \pi_j} |G[w](f)|$, for some $r \in M_G[V]$ such that $r(\pi_j) \in M_G[E]$; otherwise, $\pi_j$ is an input vertex corresponding to input $x_{\pi_j}$, then $x_{\pi_j}$ is modified to $x_{\pi_j} \prod_{f \in \pi_j} |G[w](f)| = h'_G(x_{\pi_j})$ where $h'_G$ is the absolute-valued path enumeration function.

This completes the proof. $\square$

Now we present the extension to Lemma 1:

*Proof of Lemma 1.* Invoke Lemma 5 to get a tree network $\mu'$ such that $\nu = \mu' \circ h'$. Then one can subdivide each input edges (edges that are incident to some input neuron $i_j$) into two edges connected by a neuron with linear activation. One of the resulting egde takes the weight of the old input edge; and the other is used to remove the absolute value in the definition of the (basic) path enumeration function.

More formally, for all $p \in \mathcal{P}$, recall that $i_p$ is a particular input neuron of $\mu'$ in the decomposition lemma (Lemma 1). Since $\mu'$ is a tree neural network, we there exists a distinct node $u_p$ in $\mu'$ that is

adjacent to $i_p$. Remove the edge $i_p u_p$, add a neuron $u'_p$, connect $i_p u'_p$ and $u'_p u_p$, where the weight of the former neuron is set to $\text{sgn}\left(\prod_{e \in p} w_e\right)$ and the latter to $w[\mu'](i_p u_p)$. It is straightforward to see that with $\mu$ constructed from above, $\nu = \mu \circ h$ where $h$ is the path enumeration function. $\qquad\square$

With slight modifications to the proof technique, one can show all the results for Theorem 1, Theorem 2 and Corollary 1 to the same matrix representation as presented in the paper but with the absolute signs around them.

## C  PROOF OF TRAINING INVARIANCES

**Claim 1** (Directional convergence to non-vanishing point implies stable sign). *Assumption 2 implies Assumption 1.*

*Proof.* Let $v$ be the direction that $\frac{w(t)}{\|w(t)\|_2}$ converges to. Let $\mathcal{O}$ be the orthant that $v$ lies in. Since $v$ does not have a 0 entry, the ball $\mathcal{B}$ with radius $\min_i |v_i|/2$ and center $v$ is a subset of the interior of $\mathcal{O}$. Since $\frac{w(t)}{\|w(t)\|_2}$ converges to $v$, there exists a time $T$ such that for all $s > T$, $\frac{w(s)}{\|w(s)\|_2} \in \mathcal{B}$. Thus, eventually, $w(s) \in \mathcal{B}$ where its signs stay constant. $\qquad\square$

**Claim 2** (Directional convergence does not imply stable activation). *There exists a function $w : \mathbb{R} \to \mathbb{R}^d$ such that $w(t)$ converges in direction to some vector $v$ but for all $u \in w(\mathbb{R})$, $w^{-1}(u)$ has infinitely many elements. This means that for some ReLU network empirical risk function $\mathcal{R}(w)$ whose set of nondifferentiable points $D$ has nonempty intersection with $w(\mathbb{R})$, the trajectory $\{w(t)\}_{t \geq 0}$ can cross a boundary from one activation pattern to another an infinite number of times.*

*Proof.* Fix a vector $v \in \mathbb{R}^d$. Consider the function $t \mapsto v|t \sin(t)|$. $\qquad\square$

**Lemma 8** (Clarke partial subderivatives of inputs with the same activation pattern is the same). *Assume stable sign regime (Assumption 1). Let $p_1, p_2 \in \mathcal{P}$ be paths of the same length $L$. Let $p_1 = \{v_1, \ldots, v_L\} \in V^L$ and $p_2 = \{u_1, \ldots, u_L\} \in V^L$. Assume that for each $i \in [L]$, we have $v_i$ and $u_i$ having the same activation pattern for each input training example, where the activation of a neuron is 0 if it is ReLU activated and has negative preactivation; and is 1 if it is linearly activated or has nonnegative preactivation. Then $\partial_{p_1} \mu(h) = \partial_{p_2} \mu(h)$ where $\partial$ is the Clarke subdifferential.*

*Proof.* In this proof, we use the absolute-valued version of the decomposition lemma. Fix a training example $j$ and some weight $w_0$ and let the output of the path enumeration function be $h := (h_p = h(x_j; w_0))_{p \in \mathcal{P}}$. Denote $\mathcal{X} \subseteq \mathbb{R}^2$ the input space of all possible pairs of values of $(p_1, p_2)$ such that Assumption 1 and the extra assumption that both paths have the same activation pattern on each neuron hold. Let $m : \mathbb{R}^2 \to \mathbb{R}$ be the same function as the tree network $\mu$ but with all but the two inputs at $p_1, p_2$ frozen. We will show that $m$ is symmetric in its input. Once this is establish, it is trivial to invoke the definition of the Clarke subdifferential to obtain the conclusion of the lemma.

Let $(a, b) \in \mathcal{X} \subseteq \mathbb{R}^2$. We thus show that if $(b, a) \in \mathcal{X}$ then $m(a, b) = m(b, a)$. Recall that $\mu$ is itself a ReLU-activated (in places where the corresponding original neurons are ReLU-activated) neural network. The fact that $\mu$ has a tree architecture means that for each input node, there is a unique path going to the output node. Thus, the set of paths from some input node in $\mu$ to its output node can be identified with the set of inputs itself: $\mathcal{P}$. Now let us considered the product of weights on some arbitrary path $p$. It is not hard to see that for each such path, the product is just $P_p = \prod_{e \in p} w_e$ since the input to $p$ is $|P|$ and going along the path collects all signs of $w_e$ for all $e \in p$.

We now invoke the 0-1 form of ReLU neural network to get $m(a, b) = \mu(h) = \sum_{p \in \mathcal{P}} Z_p(a, b) P_p(a, b)$ where $Z_p$ is 1 iff all neurons on path $p$ is active in the $\mu$ network (recall that we identify paths in $\mu$ to input nodes). Consider that what changes between $m(a, b)$ and $m(b, a)$: since $\mu$ is a tree network, exchanging two inputs can only have effect on the activation pattern of neurons along the two paths from these inputs. However, we restricted $\mathcal{X}$ to be the space where activation pattern of each neuron in the two paths is identical to one another. Since both $(a, b)$ and $(b, a)$ is in $\mathcal{X}$, swapping one for another does not affect the activation pattern of each neuron in the two paths! These neurons activation pattern is then identical to those in network $\mu$ by construction

and hence $Z_p(a, b) = Z_p(b, a)$ for all $p \in \mathcal{P}$ since activation pattern of each node of the $\mu$ network stays the same. Thus we conclude that $m(a, b) = m(b, a)$. This completes the proof. $\qquad \square$

*Proof of Lemma 2.* This proof uses the absolute-value-free version of the decomposition lemma. This is just to declutter notations, as the same conclusion can also be reach using the other version, with some keeping track of weight signs. Recall that our real weight vector $w(t)$ is an arc Ji & Telgarsky (2020) which, by definition, is absolutely continuous. It then follows from real analysis that its component $w_e(t)$ is also absolutely continuous for any $e \in E$. For absolutely continuous functions $w_e(t)$ for some $e \in E$, invoke the Fundamental Theorem of Calculus (FTC) (Chapter 6, (Heil, 2019)), to get:

$$\sum_{u \in \mathrm{IN}_v} w_{uv}^2(t) - \sum_{u \in \mathrm{IN}_v} w_{uv}^2(0) = 2 \sum_{u \in \mathrm{IN}_v} \int_{[0,t]} w_{uv}(s) \frac{\mathrm{d}w_{uv}}{\mathrm{d}t}(s)\mathrm{d}s \tag{30}$$

We now proceed to compute $\frac{\mathrm{d}w}{\mathrm{d}t}(s)$. Since $w(t)$ is absolutely continuous and we are taking the integral via FTC, we only need to compute $\frac{\mathrm{d}w}{\mathrm{d}t}(s)$ for a.e. time $s$. By chain rule (see for example, the first line of the proof of Lemma C.8 in Lyu & Li (2020)), there exists functions $(g_j)_{j=1}^n$ such that $g_j \in \partial \nu_{x_j}(w) \subseteq \mathbb{R}^{|E|}$ for all $j \in [n]$ where $\nu_{x_j}(w) = \nu(x_j; w)$, and for a.e. time $s \geq 0$,

$$\frac{\mathrm{d}w}{\mathrm{d}t}(s) = \frac{1}{n} \sum_{j=1}^n l'(y_j \nu(x_j; w(s))) \cdot y_j \cdot g_j \tag{31}$$

Fix $j \in [n]$, by the inclusion chain rule, since $\nu_{x_j} = \mu \circ h_{x_j}$, with $h$ and $\mu$ also locally Lipschitz, we have by Theorem I.1 of Lyu & Li (2020),

$$\partial \nu_{x_j}(w) = \partial(\mu \circ h_{x_j})(w) \subseteq \mathrm{CONV} \left\{ \sum_{p \in \rho} [\alpha]_p \beta_{j,p} \mid \alpha \in \partial\mu(h(w)), \beta_{j,p} \in \partial[h_{x_j}]_p(w) \right\}. \tag{32}$$

Thus there exists $(\gamma_a)_{a=1}^A \geq 0, \sum_{a=1}^A \gamma_a = 1$ and $(\alpha^a \in \partial\mu(h(w)), \beta_{j,p}^a \in \partial[h_{x_j}]_p(w))_{a=1}^A$ such that:

$$g_j = \sum_{a=1}^A \gamma_a \sum_{p \in \rho} [\alpha^a]_p \beta_{j,p}^a.$$

Here we use Assumption 1 to deduce that eventually, all weights are non-zero in gradient flow trajectory to compute:

$$\partial[h_{x_j}]_p(w) = \left\{ \frac{\mathrm{d}[h_{x_j}]_p(w)}{\mathrm{d}(w)} \right\} = \left\{ \left( \mathbb{1}_{e \in p} \cdot (x_j)_p \prod_{f \in p, f \neq e} w_f \right)_{e \in E} \right\}.$$

Plug this back into $g_j$ to get:

$$g_j = \left( \sum_{a=1}^A \gamma_a \sum_{p \in \rho | e \in p} [\alpha]_p^a \cdot (x_j)_p \prod_{f \in p, f \neq e} w_f \right)_{e \in E}.$$

Plug this back into $\frac{\mathrm{d}w}{\mathrm{d}t}(s)$ and to get, coordinate-wise, for a.e. $s \geq 0$,

$$\frac{\mathrm{d}w_e}{\mathrm{d}t}(s) = \frac{1}{n} \sum_{j=1}^n l'(y_i \nu(x_i; w(s))) \cdot y_i \cdot \sum_{a=1}^A \gamma_a \sum_{p \in \rho | e \in p} [\alpha^a]_p \cdot (x_j)_p \prod_{f \in p, f \neq e} w_f(s). \tag{33}$$

Multiply both sides with $w_e$ gives:

$$w_e(s) \frac{\mathrm{d}w_e}{\mathrm{d}t}(s) = \sum_{p \in \mathcal{P} | e \in p} \frac{1}{n} \sum_{j=1}^n d_{j,p}(w(s)), \tag{34}$$

where

$$d_{j,p}(w) = \ell'(y_i \nu(x_i; w)) \cdot y_i \cdot \sum_{a=1}^{A} \gamma_a [\alpha]_p^a \cdot (x_j)_p \cdot \prod_{f \in p} |w_f(s)|. \tag{35}$$

Note that $d$ does not depend on the specific edge $e$ used in Equation 33 and also that the term given by $\beta$ does not depend on $a$ and we can simply write $\alpha_p$ for $\sum_{a=1}^{A} \gamma_a [\alpha]_p^a$.

Plugging back into the FTC to get:

$$\sum_{u \in \text{IN}_v} w_{uv}^2(t) - \sum_{u \in \text{IN}_v} w_{uv}^2(0) = 2 \sum_{u \in \text{IN}_v} \int_{[0,t]} \sum_{p \in \mathcal{P} | uv \in p} \frac{1}{n} \sum_{j=1}^{n} d_{j,p}(w(s)) \mathrm{d}s \tag{36}$$

$$= 2 \int_{[0,t]} \sum_{p \in \mathcal{P} | v \in p} \frac{1}{n} \sum_{j=1}^{n} d_{j,p}(w(s)) \mathrm{d}s. \tag{37}$$

Finally, by an identical argument but applied to the set of edges $vb$ for some $b \in \text{OUT}_v$, we have:

$$\sum_{b \in \text{OUT}_v} w_{vb}^2(t) - \sum_{b \in \text{OUT}_v} w_{vb}^2(0) = 2 \int_{[0,t]} \sum_{p \in \mathcal{P} | v \in p} \frac{1}{n} \sum_{j=1}^{n} d_{j,p}(w(s)) \mathrm{d}s \tag{38}$$

$$= \sum_{u \in \text{IN}_v} w_{uv}^2(t) - \sum_{u \in \text{IN}_v} w_{uv}^2(0), \tag{39}$$

which completes the proof of the first part of the lemma.

For the second part, recall that we have 2 vertices $u, v$ such that $\text{IN}_v = \text{IN}_u = \text{IN}$ and $\text{OUT}_v = \text{OUT}_u = \text{OUT}$ with stable activation pattern for each training example. To make it more readable, we drop the explicit dependence on $t$ in our notation and introduce some new ones: for some $a \in \text{IN}$, let $\mathcal{P}_{I \to a}$ be the set of all paths from some input node in $I$ to node $a$ and for some $b \in \text{OUT}$, let $\mathcal{P}_{b \to o}$ be the set of all paths from $b$ to the output node $o$. Then one can decompose the sum as

$$\frac{d}{dt} w_{au} = \sum_{j=1}^{n} -\ell'(y\nu(x_j; w)) \cdot y \cdot \sum_{p_1 \in \mathcal{P}_{I \to a}} \sum_{b \in \text{OUT}} \sum_{p_2 \in \mathcal{P}_{b \to o}} (x_j)_{p_1} \cdot w_{ub} \cdot \prod_{f \in p_1 \cup p_2} w_f \cdot \alpha_{p_1 \cup \{u\} \cup p_2}, \tag{40}$$

where $\alpha_p$ is the partial Clarke subdifferential at input $p$.

Recall that the end goal is to derive

$$\frac{d}{dt} w_{au} w_{av} = w_{au} \frac{d}{dt} w_{av} + w_{av} \frac{d}{dt} w_{au}. \tag{41}$$

Using equation equation 40, the second term on the right hand side becomes

$$w_{av} \frac{d}{dt} w_{au} = \sum_{j=1}^{n} -\ell'(y\nu(x_j; w)) \cdot y \cdot \sum_{p_1 \in \mathcal{P}_{I \to a}} \sum_{b \in \text{OUT}} \sum_{p_2 \in \mathcal{P}_{b \to o}} \tag{42}$$

$$\left[ (x_j)_{p_1} \cdot w_{av} \cdot \left( \prod_{f \in p_1 \cup p_2} w_f \right) \cdot w_{ub} \right] \cdot \alpha_{p_1 \cup \{u\} \cup p_2} (x_j; w) \tag{43}$$

where the product $w_{av} \cdot \left( \prod_{f \in p_1 \cup p_2} \cdot w_f \right) w_{ub}$ is a jagged path. Then one continues with the derivation to get

$$\frac{d}{dt} \left( \sum_{a \in \text{IN}} w_{au} w_{av} \right) = \sum_{a \in \text{IN}} \frac{d}{dt} (w_{au} w_{av}) \tag{44}$$

$$= \sum_{j=1}^{n} -\ell'(y\nu(x_j; w)) \cdot y \cdot \sum_{a \in \text{IN}} \sum_{b \in \text{OUT}} \sum_{p_1 \in \mathcal{P}_{I \to a}} \sum_{p_2 \in \mathcal{P}_{b \to o}} \tag{45}$$

$$\left( (x_j)_{p_1} \cdot \prod_{f \in p_1 \cup p_2} w_f \right) \left( w_{av} \cdot w_{ub} \cdot \alpha_{p_1 \cup \{u\} \cup p_2}(x_j; w) + w_{au} \cdot w_{vb} \cdot \alpha_{p_1 \cup \{v\} \cup p_2}(x_j; w) \right)$$
$$\tag{46}$$

On the other hand

$$\frac{d}{dt} \left( \sum_{b \in \text{OUT}} w_{ub} w_{vb} \right) = \sum_{b \in \text{OUT}} \frac{d}{dt} (w_{ub} w_{vb}) \tag{47}$$

$$= \sum_{j=1}^{n} -\ell'(y\nu(x_j; w)) \cdot y \cdot \sum_{a \in \text{IN}} \sum_{b \in \text{OUT}} \sum_{p_1 \in \mathcal{P}_{I \to a}} \sum_{p_2 \in \mathcal{P}_{b \to o}} \tag{48}$$

$$\left( (x_j)_{p_1} \cdot \prod_{f \in p_1 \cup p_2} w_f \right) \left( w_{av} \cdot w_{ub} \cdot \alpha_{p_1 \cup \{v\} \cup p_2}(x_j, w) + w_{au} \cdot w_{vb} \cdot \alpha_{p_1 \cup \{u\} \cup p_2}(x_j; w) \right).$$
$$\tag{49}$$

This is where the more restrictive assumption that for each training example, $u$ and $v$ have the same activation pattern as each other (but the same activation pattern between $u$ and $v$ of, say $x_1$, may differs from that on, say $x_2$). Under this assumption, we can invoke Lemma 8 to get $\alpha_{p_1 \cup \{u\} \cup p_2}(x_j; w) = \alpha_{p_1 \cup \{v\} \cup p_2}(x_j; w)$ as a set. This identifies 46 with 49 and gives us:

$$\frac{d}{dt} \left( \sum_{a \in \text{IN}} w_{au}(t) w_{av}(t) - \sum_{b \in \text{OUT}} w_{ub}(t) w_{vb}(t) \right) = 0. \tag{50}$$

This holds for any time $t$ where $u$ and $v$ has the same activation pattern. If further, they have the same activation pattern throughout the training phase being considered, then one can invoke FTC to get the second conclusion of Lemma 2. Note, however, that we will be using this differential version in the proof of Theorem 1.

$\square$

*Proof of Lemma 3.* The proof is identical to that of Lemma 2, with the only difference being the set $\mathcal{A}$ that we double count. Here, set $\mathcal{A}$ to be $\mathcal{P}$. Then by the definition of layer (see Definition 7), one can double count $\mathcal{P}$ by counting paths that goes through any element of a particular layer. The proof completes by considering (using the same notation as the proof of Lemma 2) for a.e. time $s \geq 0$,

$$\sum_{e \in F} w_e(s) \frac{dw_e}{dt}(s) = \sum_{p \in \mathcal{P}} \frac{1}{n} \sum_{j=1}^{n} d_{j,p}(w(s)) \cdot (x_j)_p \cdot \prod_{f \in p} w_f(s), \tag{51}$$

for any layer $F$.

$\square$

Before continuing with the proof of Theorem 1, we state a classification of neurons in a convolutional layer. Recall that all convolutional layers in this paper is linear. Due to massive weight sharing within a convolutional layer, there are a lot more neurons and edges than the number of free parameters in this layer. Here, we formalize some concepts:

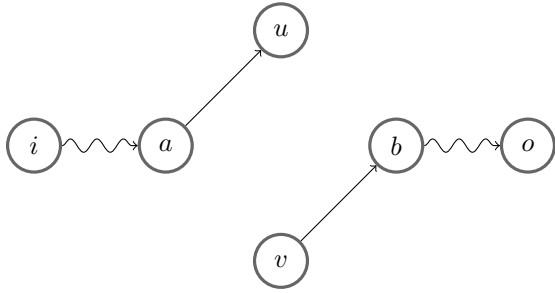

Figure 4: Jagged path. Here the straight arrow denote a single edge in the graph while the snaked arrow denote a path with possibly more than one edge. $u$ and $v$ are nodes in the statement of the second part of Lemma 2, $a \in \text{IN} = \text{IN}_v = \text{IN}_u$, $b \in \text{OUT} = \text{OUT}_v = \text{OUT}_u$.

**Definition 11** (Convolutional tensors). *Free parameters in a (bias-free) convolutional layer can be organized into a tensor $T \in R^{a \times b \times c}$ where $a$ is the number of input channels, $b$ is the size of each 2D filter and $c$ is the number of output channels.*

For example, in the first convolutional layer of AlexNet, the input is an image with 3 image channels (red, blue and green), so $a = 3$ in this layer; filters are of size $11 \times 11$ so $b = 121$ is the size of each 2D filter; and finally, there are 96 output channels, so $c = 96$. Note that changing the stride and padding does not affect the number of free parameters, but does change the number of neurons in the computation graph of this layer. Thus, we need the following definition:

**Definition 12** (Convolutional neuron organization). *Fix a convolutional layer with free parameters $T \in R^{c \times b \times a}$. Then neurons in this layer can be organized into a two dimensional array $\{v_{j,k}\}_{j \in [l], k \in [c]}$ for some $l \in \mathbb{N}$ such that*

1. *For all $j, j' \in [l]$ and for all $k \in [c]$, $v_{j,k}$ and $v_{j',k}$ share all its input weights. More formally, there exists a bijection $\phi : \text{IN}_{v_{j,k}} \to \text{IN}_{v_{j',k}}$ such that:*

$$w_{u v_{j,k}} \equiv w_{\phi(u) v_{j',k}}, \tag{52}$$

*for all $u \in \text{IN}_{v_{j_k}}$.*

2. *For all $j \in [l]$ and for all $k, k' \in [c]$, $v_{j,k}$ and $v_{j,k'}$ has the same in-vertices and out-vertices. In other words,*

$$\text{IN}_{v_{j_k}} = \text{IN}_{v_{j,k'}} =: \text{IN}_{v_j} \text{ and } \text{OUT}_{v_{j_k}} = \text{OUT}_{v_{j,k'}} =: \text{OUT}_{v_j}. \tag{53}$$

For example, in the first convolutional of AlexNet in $\mathbb{R}^{96 \times 121 \times 3}$, the input image dimension is $224 \times 224 \times 3$ pixels and the stride of the $11 \times 11 \times 3$ filters in the layer is 4, with no padding. Thus a single 2D filter traverse the image $\left( \frac{224 - (11 - 4)}{4} \right)^2 =: l$ times, each corresponds to a different neuron in the layer. This process is then repeated $c$ times for each output channel, for a total of $c \times l$ neurons in the convolutional layer.

**Lemma 9** (Extension of Lemma 2 to weight sharing). *Let $\mathcal{V} := \{v_{j,k}\}_{j \in [l], k \in \{1,2\}}$ be a convolutional neuron organization. Recall that by definition 12, for all $j \in [l]$, $\text{IN}_{v_{j,1}} = \text{IN}_{v_{j,2}} =: \text{IN}_j$ and $\text{OUT}_{v_{j,1}} = \text{OUT}_{v_{j,2}} =: \text{OUT}_j$. We have for a.e. time $t \geq 0$,*

$$\sum_{(w,w') \in w_{\text{IN}}(\mathcal{V})} |w(t)w'(t)| - |w(0)w'(0)| = \sum_{(z,z') \in w_{\text{OUT}}(\mathcal{V})} |z(t)z'(t)| - |z(0)z'(0)|, \tag{54}$$

*where*

$$w_{\text{IN}}(\mathcal{V}) := \{(w_1, w_2) \mid \forall j \in [l], \exists a_j \in \text{IN}_j, w_{a_j v_{j,1}} \equiv w_1 \text{ and } w_{a_j v_{j,2}} \equiv w_2\},$$

*and similarly,*

$$w_{\text{OUT}}(\mathcal{V}) := \{(w_1, w_2) \mid \forall j \in [l], \exists b_j \in \text{OUT}_j, w_{v_{j,1} b_j} \equiv w_1 \text{ and } w_{v_{j,2} b_j} \equiv w_2\}.$$

Before we start the proof, some remarks are in order. Let $T_1 \in \mathbb{R}^{2 \times b \times a}$ be a convolutional layer and let $\mathcal{V} = \{v_{j,k}\}_{j \in [l], k \in [2]}$ be its convolutional neuron organization. Then

$$w_{\text{IN}}(\mathcal{V}) = \{([T_1]_{1,j_2,j_1}, [T_1]_{2,j_2,j_1}) \mid j_1 \in [a], j_2 \in [b]\}. \tag{55}$$

If furthermore $T_2 \in R^{e \times d \times 2}$ is a convolutional layer immediately after $T_1$, then

$$w_{\text{OUT}}(\mathcal{V}) = \{([T_2]_{k_1,k_2,1}, [T_2]_{k_1,k_2,2}) \mid k_1 \in [e], k_2 \in [d]\}. \tag{56}$$

If instead of $T_2$, the subsequent layer to $T_1$ is a fully-connected layer $W \in \mathbb{R}^{d \times (l \times 2)}$ (recall that there are $2l$ neurons in $T_1$ layer), then

$$w_{\text{OUT}}(\mathcal{V}) = \{(W_{l_1,l_2 \times 1}, [T_2]_{l_1,l_2 \times 2}) \mid l_1 \in [d], l_2 \in [l]\}. \tag{57}$$

*Proof of Lemma 9.* As before, the proof is identical to that of Lemma 2 with the exception being the set of paths that we are double counting over. For all $j \in [l]$, let

$$\mathcal{A}_j := \{p_1 \cup p_2 \mid p_1 \text{ is a path from } I \text{ to } v_{j,1}, p_2 \text{ is a path from } v_{j,2} \text{ to } o\}, \tag{58}$$

and

$$\mathcal{A}'_j := \{p_1 \cup p_2 \mid p_1 \text{ is a path from } I \text{ to } v_{j,2}, p_2 \text{ is a path from } v_{j,1} \text{ to } o\}. \tag{59}$$

Let $\mathcal{A} := \bigcup_{j \in [l]} \mathcal{A}_j \cup \mathcal{A}'_j$. Then by an identical argument as that of Lemma 2, we can show that

$$\sum_{(w,w') \in w_{\text{IN}}(\mathcal{V})} w(t)w'(t) - w(0)w'(0) = \int_{[0,t]} \sum_{p \in \mathcal{A}} \frac{1}{n} \sum_{j=1}^{n} d_{j,p}(w(s)) \mathrm{d}s \tag{60}$$

$$= \sum_{(z,z') \in w_{\text{OUT}}(\mathcal{V})} z(t)z'(t) - z(0)z'(0). \tag{61}$$

Here the notation $d_{i,j}$ is well-defined since we do not have to specify a path for the subgradient $\alpha_p$. This is because we are working with linear convolutional layers and thus all subgradients are gradients and is evaluated to 1.

$\square$

*Proof of Theorem 1.* All points in this theorem admit the same proof technique: First, form the matrices as instructed. Let $X = \{v_1, v_2, \ldots, v_m\} \subseteq V$ be the active neurons shared between the two layers. Check the conditions of the second part of Lemma 2 and invoke the lemma for each pair $u, v \in X$. We now apply this to each points:

1. Let $W_1 \in \mathbb{R}^{b \times a}$ be a fully-connected layer from neurons in $V_1$ to neurons in $V_2$ and $W_2 \in \mathbb{R}^{c \times b}$ be a fully-connected layer from $V_2$ to $V_3$. Then we have $\text{IN}_u = V$ and $\text{OUT}_u = W$ for all $u \in U$. Furthermore, all weights around any $u$ are learnable for all $u$ in $U$. Invoke the second part of Lemma 2 to get the conclusion.

2. let $T_1 \in \mathbb{R}^{c \times b \times a}$ and $T_2 \in \mathbb{R}^{e \times d \times c}$ be the convolutional tensors with convolutional neuron organization of $T_1$ (Definition 12) being $\{v_{j,k}\}_{j \in [l_1], k \in [c]}$. Form the matrix representation $W_1 \in \mathbb{R}^{c \times (ab)}$ and $W_2 \in \mathbb{R}^{(de) \times c}$ as per the theorem statement. By Definition 12, for $k, k' \in [c]$, for all $j \in [l]$, $\text{IN}_{v_{j,k}} = \text{IN}_{v_{j,k'}}$ and $\text{OUT}_{v_{j,k}} = \text{OUT}_{v_{j,k'}}$. Invoke Lemma 9 to get the conclusion.

3. Let $r(x; U, Y, Z)$ be a residual block of either ResNetIdentity, ResNetDiagonal or ResNet-Free. In all variants, skip connection affects neither the edges in $U \in \mathbb{R}^{b \times a}$ and $Y \in \mathbb{R}^{c \times b}$. Let $Y$ be fully-connected from neurons $V_1$ to neurons $V_2$ and $U$ be fully-connected from neurons $V_2$ to neurons $V_3$. Then for each $u \in V_2$, $\text{IN}_u = V_1$ and $\text{OUT}_u = V_3$. Furthermore, all weights around any vertices in $V_2$ are learnable. Invoke the second part of Lemma 2 to get the conclusion.

4. Let $r_i(x; U_i, Y_i, Z_i), i \in \{1, 2\}$ be consecutive ResNetFree block. Let $Y_1$ be fully connected from neurons $V_1$ to neurons $V_2$, $U_1$ be fully-connected from $V_2$ to neurons $V_3$, $Y_2$ be fully-connected from neurons $V_3$ to $V_4$ and $U_2$ be fully-connected from $V_4$ to neurons $V_5$. Then $\begin{bmatrix} U_1 & Z_1 \end{bmatrix}$ is fully-connected from $V_1 \cup V_2$ to $V_3$ and $\begin{bmatrix} Y_2 \\ Z_2 \end{bmatrix}$ is fully-connected from $V_3$ to $V_4 \cup V_5$. Invoke the first point to get the conclusion.

5. Let the convolutional tensor be $T \in \mathbb{R}^{c \times b \times a}$ with convolutional neuron organization $\{v_{j,k}\}_{j \in [l], k \in [c]}$ for some $l \in \mathbb{N}$. Let the adjacent fully-connected layer be $W \in \mathbb{R}^{d \times (l \times c)}$. Form the matrix representation $W_1 \in \mathbb{R}^{c \times ab}$ and $W_2 \in \mathbb{R}^{dl \times c}$ as per the theorem statement. Then we have for any $k, k' \in [c]$ and for all $j \in [l]$, $\text{IN}_{v_{j,k}} = \text{IN}_{v_{j,k'}} =: \text{IN}_j$ and $\text{OUT}_{v_{j,k}} = \text{OUT}_{v_{j,k'}} =: \text{OUT}_j$. Invoke Lemma 9 to get the conclusion.

6. Let $r(x; U, Y, Z)$ be a residual block of either ResNetIdentity, ResNetDiagonal or ResNet-Free. Let $Y$ be fully-connected from neurons $V_1$ to neurons $V_2$, $U$ be fully-connected from neurons $V_2$ to neurons $V_3$. Thus, $Z$ is fully-connected from $V_1$ to $V_3$. Then $W_2 = \begin{bmatrix} U & Z \end{bmatrix}$ is fully connected from $V_1$ to $V_2 \cup V_3$. We invoke the fifth point to get the conclusion.

7. first consider the case where the ResNetFree block is followed by the fully-connected layer. Let $r(x; U, Y, Z)$ be the first ResNetFree block with input neurons where $Y$ fully-connects neurons $V_1$ to $V_2$ and $U$ fully connects $V_2$ to $V_3$. Then we have $\begin{bmatrix} U & Z \end{bmatrix}$ fully-connects $V_1 \cup V_2$ to $V_3$. If the subsequent layer is a fully-connected layer then invoke the first point to get the conclusion; otherwise if the subsequent layer is a ResNetFree block $r(x; U', Y', Z')$ with $Y'$ fully-connects $V_3$ to $V_4$ and $U'$ fully-connects $V_4$ to $V_5$. Then $\begin{bmatrix} Y \\ Z \end{bmatrix}$ fully-connects $V_3$ to $V_4 \cup V_5$ and we one again invoke the first point to get the conclusion.

$\square$

*Proof of Lemma 4.* This is the continuation of the proof of Lemma 2. To obtain noninvariance, one only needs to show that when $u$ is active and $v$ inactive, the expression in 46 and 49 are not equal in general. For the sake of notation, we pick the case where the preactivation of $u$ is strictly positive, that of $v$ is strictly negative, and further assume that the whole network is differentiable at the current weights $w$ for all training examples.

In this case, it is not hard to see that the Clarke subdifferential $\partial_w \mu(h)$ is a singleton and contains the gradient of the $\mu$ network. Furthermore, for any path $p = (v_1, \ldots, v_L)$, the partial derivative $\frac{\partial \mu}{\partial p}$ is 1 if all neurons on $p$ are active and 0 otherwise. Thus, we have

$$\frac{d}{dt}\left(\sum_{a \in \text{IN}} w_{au} w_{av}\right) \tag{62}$$

$$= \sum_{j=1}^{n} -\ell'(y\nu(x_j; w)) \cdot y \cdot \sum_{a \in \text{IN}} \sum_{b \in \text{OUT}} \sum_{\text{active } p_1 \in \mathcal{P}_{I \to a}} \sum_{\text{active } p_2 \in \mathcal{P}_{b \to o}} \left((x_j)_{p_1} \cdot \prod_{f \in p_1 \cup p_2} w_f\right) w_{av} \cdot w_{ub}. \tag{63}$$

We can actually factorize this even further by noticing that the term $w_{av}$ does not depend on $b$ and $w_{ub}$ does not depend on $a$. Rearranging the sum and factorizes give:

$$\frac{d}{dt}\left(\sum_{a \in \text{IN}} w_{au} w_{av}\right) \tag{64}$$

$$= \sum_{j=1}^{n} -\ell'(y\nu(x_j; w)) \cdot y \cdot \left(\sum_{\text{active } p_1 \in \mathcal{P}_{I \to v}} (x_j)_{p_1} \cdot \prod_{f \in p_1} w_f\right)\left(\sum_{\text{active } p_2 \in \mathcal{P}_{u \to o}} \prod_{f \in p_2} w_f\right). \tag{65}$$

On the other hand

$$\frac{d}{dt}\left(\sum_{b\in\text{OUT}} w_{ub}w_{vb}\right) \tag{66}$$

$$=\sum_{j=1}^{n} -\ell'(y\nu(x_j;w))\cdot y\cdot\left(\sum_{\text{active } p_1\in\mathcal{P}_{I\to u}} (x_j)_{p_1}\cdot\prod_{f\in p_1} w_f\right)\left(\sum_{\text{active } p_2\in\mathcal{P}_{v\to o}}\prod_{f\in p_2} w_f\right). \tag{67}$$

Take, for example, an asymmetric case where the in-edges of $v$ has much larger weights than that of $u$ while out-edges of $v$ has much smaller weights than that of $u$, then 65 is much larger than 67 and therefore the two expressions are not equal in the general case. A symmetric initialization scheme that prevents the above asymmetry may prevent this from happens, but this requires additional assumptions and is opened to future work. $\qquad\square$

**Remark 1.** *Using the automatic differentiation framework while setting the gradient of ReLU to be* 1 *if the preactivation is nonnegative while* 0 *otherwise, the same derivation of* 65 *can be achieved. Interestingly, if one only has a single example, then the final expression of* 65 *implies that the matrix* $\frac{d}{dt}\left(W_k^\top W_k\right)$ *has rank at most* 2, *where* $W_k$ *is a weight matrix in, for example, ReLU fully-connected neural network. Controlling the eigenvalues under low-rank updates may allow us to bound singular values of the full weight matrices* $W_k$. *The resulting bound would not be uniform over time, but improves with training and is thus a different kind of low rank result. This, however, is outside of the scope of this paper.*

## D PROOF OF THEOREM 2

First we state a helper lemma

**Lemma 10** (Largest singular value of different flattening of the same tensor is close)**.** *Let* $T\in\mathbb{R}^{a,b,c}$ *be an order* 3 *tensor (say a convolutional weight tensor). Let* $T_1, T_2$ *be the standard flattening of this tensor into an element in* $\mathbb{R}^{c\times(a\times b)}$ *and* $\mathbb{R}^{(b\times c)\times a}$ *respectively. Then,*

$$\frac{1}{\min(a,b)}\|T_1\|_2^2 \le \|T_2\|_2^2. \tag{68}$$

*Proof.* Invoke Theorem 4.8 of Wang et al. (2017) and using the same notation in the same paper, we have for $\pi_1 = \{\{c\},\{a,b\}\}$ and $\pi_2 = \{\{b,c\},\{a\}\}$,

$$\frac{\dim_T(\pi_1,\pi_2)}{\dim(T)}\|T_1\|_2^2 \le \|T_2\|_2^2. \tag{69}$$

All that is left is to compute the left hand side in term of $a, b, c$. By definition, $\dim(T) = abc$ and

$$\dim_T(\pi_1,\pi_2) = \dim_T(\{\{c\},\{a,b\}\},\{\{b,c\},\{a\}\}) \tag{70}$$

$$= \left[\max\left(D_T(\{c\},\{b,c\}), D_T(\{c\},\{a\})\right)\right] \tag{71}$$

$$\cdot \left[\max\left(D_T(\{a,b\},\{b,c\}), D_T(\{a,b\},\{a\}))\right)\right] \tag{72}$$

$$= \left[\max(c,0)\right]\cdot\left[\max(a,b)\right] = c\max(a,b). \tag{73}$$

Plug this in equation 69 to get the final result. $\qquad\square$

**Lemma 11** (Shuffling layer in linear ResNetFree block preserves largest singular value up to multiple by 8)**.** *Recall that for a ResNetFree block* $r(U, Y, Z)$ *with* $Y\in\mathbb{R}^{b\times a}$, $U\in\mathbb{R}^{c\times b}$ *and* $Z\in\mathbb{R}^{c\times a}$, *there are two possible rearrangement of the weights* $A = \begin{bmatrix} U & Z \end{bmatrix}$ *and* $B = \begin{bmatrix} Y \\ Z \end{bmatrix}$. *We have* $\|B\|_2^2 \ge \frac{1}{8}\|A\|_2^2 - D'$ *where* $D' \ge 0$ *is fixed at inialization.*

*Proof.* Recall that by point three of Theorem 1, we have matrix invariance

$$U^\top(t)U(t) - Y(t)Y^\top(t) = U^\top(0)U(0) - Y(0)Y^\top(0).$$

Note that we can obtain this form since we are only considering linear ResNetFree blocks, so all neurons are active at all time for all training examples. Thus, we can invoke a FTC to get this form from the differential from in theorem 1.

By line $B.2$ in Ji & Telgarsky (2019),

$$\|Y\|_2^2 \geq \|U\|_2^2 - D, \tag{74}$$

where $D = \|U^\top(0)U(0) - Y(0)Y^\top(0)\|_2^2$ is fixed at initialization.

For positive semidefinite matrix $X$, denote $\lambda(X)$ to be the function that returns the maximum eigenvalue of $X$. We have

$$\|B\|_2^2 = (\lambda(B^\top B))^2 \tag{75}$$

$$= \left(\lambda\left(Y^\top Y + Z^\top Z\right)\right)^2 \tag{76}$$

$$\geq \frac{1}{4}\left(\lambda\left(Y^\top Y\right) + \lambda\left(Z^\top Z\right)\right)^2 \tag{77}$$

$$\geq \frac{1}{4}\left(\lambda\left(Y^\top Y\right)\right)^2 + \frac{1}{4}\left(\lambda\left(Z^\top Z\right)\right)^2 \tag{78}$$

$$\geq \frac{1}{4}\left(\lambda\left(UU^\top\right)\right)^2 + \frac{1}{4}\left(\lambda\left(ZZ^\top\right)\right)^2 - \frac{D}{4} \tag{79}$$

$$\geq \frac{1}{8}\left(\lambda\left(UU^\top\right) + \lambda\left(ZZ^\top\right)\right)^2 - \frac{D}{4} \tag{80}$$

$$\geq \frac{1}{8}\left(\lambda\left(UU^\top + ZZ^\top\right)\right)^2 - \frac{D}{4} \tag{81}$$

$$= \frac{1}{8}\left(\lambda\left(AA^\top\right)\right)^2 - \frac{D}{4} = \frac{1}{8}\|A\|_2^2 - \frac{D}{2}, \tag{82}$$

where 77 is by an application of Weyl's inequality for Hermitian matrices which states that $\lambda(C + D) \geq \lambda(C) + t$ where $t$ is is the smallest eigenvalue of $D$, which is nonnegative since matrices here are all positive semidefinite; 79 is a consequence of 74 and 81 is the application of the inequality $\lambda(C + D) \leq \lambda(C) + \lambda(D)$ which is another of Weyl's inequality for Hermitian matrices. $\qquad\square$

*Proof of Theorem 2.* Fix $j \in [K + M]$, we first invoke Lemma 3 to bound the Frobenius norm of each of the final $K + M + 1$ layers (counting the last layer $Fin$) via the last layer. Let $W_j$ be the matrix representation of the $j$-th layer among the last $M + 1$ layer as described in Theorem 1. Note that even if a layer has more than one matrix representation in Theorem 1, their Frobenius norm is still the same because different representation merely re-organize the weights. Thus, we can pick an arbitrary representation in this step. However, the same is not true for the operator norm and we have to be a lot more careful in the next step. For each $j \in [K + M]$, we have

$$\|W_j(t)\|_F^2 - \|Fin(t)\|_F^2 = D_0, \tag{83}$$

where $D_0 = \|W_j(0)\|_F^2 - \|Fin(0)\|_F^2$ fixed at initialization.

Now, we bound the difference between the operator norm of $W_j(t)$ and $Fin(t)$ by a telescoping argument. By Lemma 11, switching from one matrix representation to the other for ResNet incurs at most a multiplicative factor of 8 and an additive factor cost that depends only on the initialization. In each adjacent layer, the maximum number of switch between representation is one (so that it fits the form prescribed in Theorem 1). By matrix invariance between adjacent layers of the $K + M$ pairs of adjacent layers,

$$\|W_l(t)\|_2^2 \geq C\|W_{l+1}(t)\|_2^2 - D_l, \tag{84}$$

for $C = 1/8$ if the $k + 1$ layer is a ResNetFree block (Lemma 11), $C = 1/\min(a_{k+1}, b_{k+1})$ if the $k + 1$ layer is a convolutional layer, $C = \max \dim W^{[M+1]}$ if $k = M$ (Lemma 10) and $C = 1$ otherwise; for $D_l = \|W_{l+1}^\top(0)W_{l+1}(0) - W_l(0)W_l^\top(0)\|_2^2$.

Telescope the sum and subtract from 83 to get the statement of the theorem. When Frobenius norm diverges, divide both sides by the Frobenius norm to get the ratio statement. Note that the bound $\|W\|_F^2/\text{rank}(W) \le \|W\|_2^2$ is trivial since the largest singular value squared is at least the average singular value squared.

When the lower bound of 14 is 1, it matches the upper bound and thus the largest singular value dominates all other singular values. Alignment follows from the proof of Lemma 2.6 second point in Ji & Telgarsky (2019). □

## E PROOF OF COROLLARY 1

*Proof.* Let $L$ be the number of layers in the network. Under the conditions stated in Corollary 1, Lyu & Li (2020) and Ji & Telgarsky (2020) showed that $\|w(t)\|_2$ diverges. Invoke Lemma 3 for all $k \in [L-1]$ and sum up the results, we have $L\|W_L(t)\|_F^2 = D'' + \sum_{j=1}^L \|W_j(t)\|_F^2 \|W_k\|_F^2 = D'' + \|w(t)\|_2^2$ where $D''$ is constant in $t$. Thus $\|W_j(t)\|_F^2$ diverges for all $j \in [L]$. Since the sum of all but the largest singular value is bounded, but the sum of all singular values diverge, we conclude that the largest singular value eventually dominates the remaining singular values. Together with convergence in direction for these architecture Ji & Telgarsky (2020), we have each matrix converges to its rank-1 approximation.

That all the feedforward neural networks with ReLU/leaky ReLU/linear activations can be definable in the same o-minimal structure that contains the exponential function follows from the work of Ji & Telgarsky (2020) and that definability is closed under function composition. □

