# OpenReview forum: "Training invariances and the low-rank phenomenon: beyond linear networks"
_ICLR.cc/2022/Conference — ICLR 2022 Spotlight_

### Official Review · Reviewer_udhX · 2021-10-28

**Correctness:** 4
**Technical Novelty And Significance:** 4
**Empirical Novelty And Significance:** 3
**Recommendation:** 8
**Confidence:** 4

**Main Review:**

************************************************************************************
Update: Thanks to the author for the response. I think now the paper is clear, and I have raised my score.
************************************************************************************

This paper is mostly well written. In my opinion, the training invariance is very relevant in theoretically understanding the deep neural networks and the results presented in this paper are novel and interesting.

However, I have questions about some statements in the paper. I hope the author can clarify the following:
1. The author stated, after Theorem 1, "The first point of Theorem 1 admits an extremely simple proof for the linear fully-connected network case in Arora et al. (2018) (Theorem 1) but has not been generalized to other architectures." I don't see why this is true: In Theorem 1, if I understand the notation correctly, equation 6. shows the invariance of $|W_1(t)|^T|W_1(t)|-|W_2(t)||W_2(t)|^T$, where $|W_1(t)|$ is the entry-wise absolute value of $W_1(t)$. However, Arora et al. (2018) show the invariance of $W_1(t)^TW_1(t)-W_2(t)W_2(t)^T$, when using linear activation. The two matrices shown to be invariant are different.
2. In the proof of Lemma 6, the non-absolute version of the invariance $W_1(t)^TW_1(t)-W_2(t)W_2(t)^T$ is used, I understand that Lemma 6 is true for linear networks because that matrix is invariant, but then Lemma 6 would not hold for nonlinear networks since now the matrix invariance has entry-wise absolute value?

Overall, I hope the author can precisely define what is the matrix invariance for nonlinear networks, and how such invariance is used to prove Theorem 2.

Minor comment: In equation (14), the inequality should only hold when taking the limit?

**Summary Of The Paper:**

This paper studies the invariance in training nonlinear networks under gradient flow or differential inclusion. It is shown that, for various network architectures, there are time-invariant quantities related to networks weights, if the trajectory is in the sign-stable regime (the sign of the weights does not change). This sign-stable regime invariance suggests the weight matrices are necessarily low-rank, provided that they diverge to infinity along some non-degenerate directions.

**Summary Of The Review:**

Strength: Well written; Novel and significant results

Weakness: Some statements might be wrong; Some proof needs clarification

---

> ### Author Response · Authors · 2021-11-17
> **Response to Reviewer udhX**
>
> We thank the reviewer for their time and comments.
>
> We now address the questions raised.
>
> 1. Absolute value of weights:
>
>     Indeed, in the submitted paper, our invariances only hold for the entry-wise
> absolute absolute value of the weight matrices and therefore, our comment that
> Arora et al., (2018) showed this for linear networks is incorrect. However,
> since then, we have been able to strengthen the proof to allow for invariance
> of both the entry-wise absolute value of the weight matrices and the actual
> weight matrix itself (which is exactly the version proven by Arora et al.,
> (2018) for linear networks). We hope that this stronger result addresses your
> concern.
>
> 2. Lemma 6:
>
>     Indeed, in the submitted paper, there was an error in the statement of Lemma 6: $A$ and $B$ should have been defined as the absolute value of their corresponding definition in that version. That $A$ and $B$ is in absolute value is also necessary for Lemma 6 to be used to prove Theorem 2. The proof still follows since we invoke our Theorem 1 to get matrix-wise invariance for nonlinear networks; and line B.2 in Ji & Telgarsky, (2019) holds as long as matrix-wise invariance holds. However, we believe that the stronger results in the revised version of the paper can address this concern since we now also have Theorem 1 for the actual weight matrices and not just its absolute values.
>
> Again, we thank the reviewer for their careful revision and helpful comments and we hope to hear back from them if there are further questions or comments.

---

### Official Review · Reviewer_eeoo · 2021-10-30

**Correctness:** 4
**Technical Novelty And Significance:** 3
**Empirical Novelty And Significance:** Not applicable
**Recommendation:** 6
**Confidence:** 3

**Main Review:**

This paper has several strengths: (1) The decomposition of any feedforward network into a tree network and a path enumeration mapping. This lemma is interesting and may be useful in future works. (2) Based on certain assumptions, the training invariance lemma, and the low-rank phenomenon can be extended to non-linear networks, which can be viewed as a great contribution. (3) The paper is well written and easy to follow.

Weakness:

(1) My main concern is that Assumption 1 and Assumption 2 are restrictive and even a bit unrealistic for some architectures. Assumption 2 seems to require the weights to go to infinity. This only happens in classification tasks, and only makes sense for homogeneous networks. For a non-homogeneous network architecture, scaling up each entry of the weight can in fact drastically change the correctness of the prediction. For example, the weights of a ResNet cannot enjoy a directional convergence since scaling up will alter the sign of the output. While the authors admit this and argue '...is widely used in practice:  most practical stopping criteria based on parameter convergence imply at least directional convergence', I do wish to see some citations to support this claim.

(2) Assumption 1 certainly cannot happen for any $t>0$. It definitely can hold for those $t$ in the late phase of training. Therefore, it would be better to instead state $t>T$ for some $T$, which is a more realistic assumption. Again I am very willing to change my mind if I see any empirical evidence supporting Assumption 1($\forall t>0$).

(3) Along with the last concern, another concern is that Assumption 1 gives an impression that the key difficulties are assumed away. The invariance lemma is proved for various architectures, but almost none of those architectures can be expected to satisfy Assumption 1 for any $t>0$. If Assumption 1 is modified so that $t>T$, then the results become less impressive since the results probably only hold for the late training phase.

**Summary Of The Paper:**

This paper extends several technical results: (1) the decomposition lemma that applies to any feedforward networks; (2) the training invariance lemma, both for vertices and layers, that applies to various network structures; (3) the low-rank phenomenon for non-linear, non-homogeneous networks.

**Summary Of The Review:**

A very interesting paper that has certain technical contributions. The assumptions may be a bit unrealistic, but the technical results can be applied in the future. I recommend acceptance and am willing to raise scores given that my concerns are adequately addressed.

---

> ### Author Response · Authors · 2021-11-17
> **Response to Reviewer eeoo**
>
> We thank the reviewer for their time and comments.
>
> We now address the questions and concerns raised by the reviewer.
>
> 1. Are the assumptions too restrictive and not likely to hold for general architectures?
>
>     - We agree that Assumption 2 is relatively restrictive, and that it is not likely to hold for ResNet. However, we emphasize that all main results (Theorem 1, 2) do not assume Assumption 2 but only Assumption 1, which is weaker and is implied by Assumption 2. In particular, we believe that it is not unreasonable to expect weight signs to stay constant for ResNet, eventually, even without directional convergence.
>
>     - Assumption 2 does not assume that the weights diverge by itself, since pointwise convergence implies directional convergence. However, we did make an explicit assumption on the divergence of the weights in Equation 14 and 15 of Theorem 2, which makes sense in a classification setting with certain losses such as the logistic loss. This is in line with previous works in the literature, for example Lyu & Li, (2020), Ji & Telgarsky (2019, 2020), to name a few.
>
>    -  Regarding the reviewer comment on our claim that most realistic stopping criteria based on parameter convergence assumes at least parameter directional convergence, we have decided to remove that sentence to avoid confusion. The statement was originally intended to be a justification for Assumption 2, which, as the reviewer pointed out, is too restrictive for general architectures anyway and not needed for Theorems 1 and 2 as we mention above.
>
>
>
> 2. Assumption 1 and t>0:
>
>     We agree with the reviewer that if one uses a standard initialization technique to start training the neural network from scratch, Assumption 1 is not likely to hold throughout training. However, we can imagine stopping gradient flow just before the first sign change in the weights. Then all finite time results (Theorem 1, equation 13 of Theorem 2) hold for this run. We have revised the statement of Assumption 1 to make it clearer that one can stop gradient flow early to obtain finite time results. Moreover, we also agree with the reviewer that if Assumption 1 holds for all $t > T$ for some $T > 0$, then we can start the gradient flow at time $T$ and look at late stage training (which is the main setting of (Ji & Telgarsky, 2019) and (Ji & Telgarsky, 2020)). There, all finite time results and limiting results hold!
>
> 3. We hope the revised version of Assumption 1 makes it clear that we can get finite time results if we only consider gradient flow in a time period $(t_1, t_2)$ where Assumption 1 does hold. We would also like to emphasize that even in the late training phase, how one can analyze gradient flow trajectory over non-linear neural networks has been an open theoretical question. Even in time intervals when Assumption 1 does hold, activation patterns can still change drastically in a ReLU network and therefore the network is still highly nonlinear. In all, we do agree with the reviewer that Assumption 1 is not ideal for finite time guarantees and we hope to relax it further in future work.
>
> Again, we deeply thank the reviewers for their insightful feedback and we hope to hear more from them if there are still lingering questions and/or concerns.

---

> ### Comment · Reviewer_eeoo · 2021-11-29
> **Post-rebuttal Update**
>
> The authors' response has addressed my concerns. I recommend accepting this submission.

---

### Official Review · Reviewer_p57R · 2021-11-02

**Correctness:** 4
**Technical Novelty And Significance:** 3
**Empirical Novelty And Significance:** Not applicable
**Recommendation:** 8
**Confidence:** 4

**Main Review:**

This paper gives a nice overview of the techniques used and improves on them in comparison to previous work. The sketch of proofs are quite detailed. This a quite dense paper, with many definitions (for all architectures studied) but I found it clear and well written.

Some of the assumption are quite strong, such as the sign remaining fixed and the alignment to parameters with no zero weight, but at least the authors take the time to discuss these assumptions.

The final result (the convergence to a rank one solution) is new for the deep non-linear case.

**Summary Of The Paper:**

The paper shows under certain assumptions, there exist invariants in DNNs with ReLU (or leaky ReLU and identity) activations - similar to the invariants observed in linear networks. To show this invariance a decomposition of the DNN as the composition of a multilinear function and a network with {+1,-1} weights. Using these invariants they show that for linearly separable data, the absolute value of the weight matrices converge to rank one matrices (assuming directional convergence).

**Summary Of The Review:**

The article proves a new result on the convergence to a rank one solution, in some quite specific setting and under some strong assumption. The techniques used are improvements over previous techniques, they are well presented and could be of independent interest.

---

> ### Author Response · Authors · 2021-11-17
> **Response to Reviewer p57R**
>
> We thank the reviewer for their positive comments.

---

### Official Review · Reviewer_6P7z · 2021-11-03

**Correctness:** 4
**Technical Novelty And Significance:** 4
**Empirical Novelty And Significance:** 4
**Recommendation:** 8
**Confidence:** 3

**Main Review:**

## Strengths
The theoretical result seems significant and to the interest of the community
The technical lemmas (Lemma 2,3,4) are well explained. Overall the paper is well written in my opinion.
## Weaknesses
I think this paper would benefit from a more detailed discussion of the interpretation of Theorem 2. (see my questions)

## Questions:
- Corollary 1: Aren’t the fact that the weight vectors do converge in direction to a limiting direction with non 0 entries implied by Assumption 1?
- I would love to see more comments and discussion around Theorem 2: To what extent (13) shows a low-rank phenomenon? What are the impact of $V_c(j)$ and $N_r(j)$ on your bound (it seems that they hurt the bound)? What is the interpretation of (14) and (15)?
- In (15) why can we look at the inner product between $v_{j+1}$ and $u_j$ (can't they have different dimensions)? What is the interpretation of this result?
- “In fact, the following assumption, shown in the literature, is sufficient” What do you mean by shown in the literature? Empirically shown? Do you have a citation for that?
- Assumption 2 implies Assumption 1 only for $t \geq t_0$. What happens if the sign for $w(t)$ changes a finite number of times and is eventually constant? Could you prove a similar result as Thm 2?

## Minor remarks
- ReLU and Relu are inconsistent across the text
- End of Page 4 ‘This assumption is realistic…’ I guess you are talking about assumption 1. Maybe, there is a missing sentence.
- Btw Equation 4 and 5 $OUT_v$. Also $u$ is not quantified for equation 5.
- Pae 9 “Phenonmenon”.
- Equation 3, Aren’t $e \in p$ missing in the index of the sum? Also, $f$ is missing in the index of the product.


**Summary Of The Paper:**

This paper studies the low-rank phenomena in the context of *non*linear neural networks. In this work, the authors prove, via a tree network transformation of their initial network, that there exists some matrix invariance across consecutive layers  (thm 1). Their assumptions are relatively mild (except maybe Assumption 1 that I discuss in my main review)

They then leverage that invariance results in order to show a low-rank phenomenon (thm 2) for a wide range of neural networks with positively-1-homogeneous activation functions.


**Summary Of The Review:**

The theoretical result shown is very relevant to the community and is proved under relatively weak assumptions.
I strongly recommend accepting this paper.

---

> ### Author Response · Authors · 2021-11-17
> **Response to Reviewer 6P7z**
>
> We thank the reviewer for their time and comments.
>
> We have fixed all the typos in `Minor remarks' and now address the questions raised:
>
> 1. Is the directional convergence assumption in Corollary 1 implied by Assumption 1?
> Indeed, that is correct. However, Assumption 1 contains both the assumption on directional convergence (that the limit exists) and the special property of the limit: no 0 entries. We did not need the full statement of Assumption 1 for this corollary because a previous work by Ji & Telgarsky (2020) has shown that under the remaining assumptions of Corollary 1, directional convergence is guaranteed in the training limit. We have revised the statement of Corollary 1 to make this clearer.
>
> 2. Discussion of Theorem 2:
> We have added a discussion of Theorem 2 and Corollary 1. In particular, we have pointed out our quantification of the low-rank phenomenon and informally stated conditions where our bound is strictly better than the trivial bound. We have also described how we obtain a rank-1 theorem in the limit for certain architectures (Corollary 1).
>
> 3. Inner product of $v_{j+1}$ and $u_j$:
> In our notation, $v_{j+1}$ (defined in the statement of Theorem 2) is the first right singular vector of $W_{k+1} \in \mathbb{R}^{n_{k+1} \times n_k}$ (defined in Section 2). Therefore $v_{j+1} \in \mathbb{R}^{n_k}$. Similarly, $u_j$ is the first left singular vector of the weight matrix $W_k \in \mathbb{R}^{n_k \times n_{k-1}}$ and thus also has $n_k$ entries and we indeed can look at their inner product. The new discussion section for Theorem 2 will cover the significance of Equation 15 as well.
>
> 4. Has Assumption 2 been shown in literature?
> We have revised the statement between Assumption 1 and Assumption 2 to make clear what we mean by "has been shown in literature". In particular, the existence and finiteness of the limit has been proven by Ji & Telgarsky (2020) for homogeneous networks under mild assumptions. A detailed discussion of this assumption in previous works is given in the next paragraph of 'Motivation and justification'
>
> 5. What if the sign of w(t) changes a finite number of times?
> Indeed, we can start the gradient flow at the point $t_0$ given by the statement of Assumption 2, in which case all finite time results (Theorem 1, equation 13 of Theorem 2) and limiting results (equation 14, 15 of Theorem 2, Corollary 1) hold. More generally, we can start the gradient flow at any point and stop just before any weight sign changes, then in that run, all finite time results hold.

---

> > ### Comment · Reviewer_6P7z · 2021-11-29
> > **Thank you for your answer**
> >
> > Thank you for your answer and the edits you made on the revision. I maintain my score. I think this paper should be accepted.

---

### Author Response · Authors · 2021-11-17
**Overview of our Revision**

We have uploaded a revision of our paper:
- We have fixed certain typos pointed out by our reviewers.
- In the main text, we added in a discussion of the significance of Theorem 2 and Corollary 1, following the review of reviewer 6P7z.
- We added in a new result, in which all our main results (Lemma 1, Theorem 1, 2, Corollary 1), now holds even without the absolute value around the weight matrices. This is inline with the form of invariance shown throughout literature without the absolute  values (for example in Arora, et al., (2018), Du, et al., (2018), etc.). Note that the version we presented with the absolute value is still correct and is detailed in the appendix but we opt to put the absolute-value-free version in the main text in order to be consistent with existing literature.
- In the appendix, we cleaned up the proofs of most of our results to be more presentable (the proof idea remains the same) and added some figures to aid the understanding of our constructions.
- In the appendix, we added straightforward extension of our gradient flow results to gradient descent under strong assumptions (smoothness of the risk), using existing techniques in Ji & Telgarsky, (2019).
- We have made minor revisions of different statements and claims following the reviews (the details of which is highlighted in our response to each individual reviewers)

---

### Public Comment · ~Thien_Le1 · 2022-04-14
**Overview of an error in the proof, the fix and how it affects results in the papers.**

Between acceptance and the final version of the paper, an error in one of the proofs surfaced, and addressing it led to modifications of the paper (in agreement with the conference chairs). Here, we summarize the changes.

In the submitted version, we described a number of invariances that should hold throughout the training of ReLU activated feedforward neural networks, namely two cases of vertex-wise invariances (Lemma 3), an edge-wise invariance (Lemma 4) and a matrix-wise invariance (Theorem 1). In the proof of these results, Kaifeng Lyu discovered a bug that invalidates one of the vertex-wise invariances (Equation 5 of Lemma 3 in the old version) and, by extension, parts of the matrix-wise invariance (Theorem 1) and parts of the low-rank results (Theorem 2) in the form they were stated originally.


In light of these, the originally stated results were modified as follows:
1. Lemma 2, Equation 4 of Lemma 3 and Lemma 4: unaffected.
2. Equation 5 of Lemma 3: needs an additional assumption that u and v have the same activation pattern for each training example.
3. Theorem 1: holds for linear CNN, linear ResNet variants, all linear fully-connected layers and submatrices of ReLU fully-connected layers containing neurons that have the same activation pattern for each training example. (There can be nonlinearities around these parts.)
4. Theorem 2 and Corollary 1: hold with the same architectures as Theorem 1.

Since we only require additional local properties for Equation 5, Theorem 1, Theorem 2 and Corollary 1, the results are still applicable to highly nonlinear and nonhomogeneous networks. For example, the additional conditions in Theorem 2 are local and confined to the last few layers. The rest of the neural network can be arbitrarily nonlinear and nonhomogeneous as long as it is feedforward.

Addressing the error has also uncovered some new results (Lemma 4, non-invariance of general ReLU layers) that are suggestive of promising future directions (see discussion after Lemma 4 and Remark 1 in Appendix C).

---

### Decision · Program_Chairs · 2022-01-20

**Decision:**

Accept (Poster)

**Comment:**

*Summary:* Low-rank bias in nonlinear architectures.

*Strengths:*
- Significant theoretical contribution.
- Well written; detailed sketch of proofs.

*Weaknesses:*
- More intuitions desired.
- Restrictive assumptions.

*Discussion:*

Authors made efforts to improve the discussion in response to 6P7z. Authors agree with eeoo about Assumption 2 being relatively restrictive but point out that main results do not need it. They discuss Assumption 1 and revised it formulation. Reviewer eeoo was satisfied with this. Following the discussion udhX raised their score (after authors acknowledged an early problems and improved them) and found the paper well written with novel and significant results.

*Conclusion:*

Three reviewers consider this a good paper that should be accepted. A fourth reviewer rated it marginally above the acceptance threshold but following the discussion period explicitly recommended acceptance. I find the topic interesting, timely, relevant. In view of unanimously favorable feedback from four reviewers I am recommending accept.